# A negative feedback loop between JNK-associated leucine zipper protein and TGF-β1 regulates kidney fibrosis

Qi Yan [1,2,6], Kai Zhu[1,6], Lu Zhang[1,3,6], Qiang Fu[1], Zhaowei Chen[1], Shan Liu[1], Dou Fu[1], Ryota Nakazato[4], Katsuji Yoshioka[4], Bo Diao[5], Guohua Ding[1], Xiaogang Li[3] & Huiming Wang [1✉]

Renal fibrosis is controlled by profibrotic and antifibrotic forces. Exploring anti-fibrosis factors and mechanisms is an attractive strategy to prevent organ failure. Here we identified the JNK-associated leucine zipper protein (JLP) as a potential endogenous antifibrotic factor. JLP, predominantly expressed in renal tubular epithelial cells (TECs) in normal human or mouse kidneys, was downregulated in fibrotic kidneys. *Jlp* deficiency resulted in more severe renal fibrosis in unilateral ureteral obstruction (UUO) mice, while renal fibrosis resistance was observed in TECs-specific transgenic *Jlp* mice. JLP executes its protective role in renal fibrosis via negatively regulating TGF-β1 expression and autophagy, and the profibrotic effects of ECM production, epithelial-to-mesenchymal transition (EMT), apoptosis and cell cycle arrest in TECs. We further found that TGF-β1 and FGF-2 could negatively regulate the expression of JLP. Our study suggests that JLP plays a central role in renal fibrosis via its negative crosstalk with the profibrotic factor, TGF-β1.

[1] Department of Nephrology, Renmin Hospital of Wuhan University, Wuhan, China. [2] Department of Geriatrics, Tongji Hospital, Tongji Medical College, Huazhong University of Science and Technology, Wuhan, China. [3] Department of Internal Medicine, and Biochemistry and Molecular Biology, Mayo Clinic, Rochester, MN 55905, USA. [4] Cancer Research Institute, Kanazawa University, Kakuma-machi, Kanazawa 920-1192, Japan. [5] Department of Medical Laboratory Center, General Hospital of Central Theater Command, Wuhan, China. [6]These authors contributed equally: Qi Yan, Kai Zhu, Lu Zhang. ✉email: WHM-renal@whu.edu.cn

Chronic kidney diseases (CKD), epidemic with high prevalence rate and high risk of progression to renal function failure, is emerging to mount a serious challenge worldwide. Kidney interstitial fibrosis, the common pathologic lesion in end stage of CKD, is characterized by interstitial inflammation, myofibroblasts proliferation, extra cellular matrix (ECM) accumulation, and loss of interstitial capillary integrity. Renal fibrosis involves large arrays of cell types and molecules, in which renal tubular epithelial cells (TECs) and transforming growth factor β (TGF-β) have drawn more attentions and been extensively studied during past decades. It is widely accepted that TECs act as early responders to injury and later as victims of fibrosis due to the impaired regenerative abilities[1,2], and TGF-β is the major driver that trigger numerous profibrotic signals[3]. TGF-β is synthesized by both tubular and interstitial cells and stimulates various cellular target responses, in particularly, in fibroblasts and myofibroblasts. Of the three TGF-β isoforms (TGF-β1, TGF-β2, and TGF-β3), TGF-β1 is well recognized as the predominant isoform in controlling renal fibrosis development[4]. The activation of TGF-β1 exerts pleiotropic cellular responses, including cell growth, differentiation, apoptosis, ECM synthesis, and autophagy, leading to the progression of renal fibrosis[3,4].

Since TGF-β1 is considered as the central mediator in renal fibrosis, large efforts have been made to explore the anti-TGF-β1 approach in renal fibrosis therapy. A significant body of evidence from the preclinical studies supports the rationality of preventing renal fibrosis and CKD progression through targeting TGF-β1 signaling pathways[5–9]. However, results from clinical trials of targeting TGF-β1 therapy are far from anticipated, and none of those TGF-β1 blockade treatments has been successfully translated into human therapy[4], which results in the urgent need to identify the other key factors in regulating renal fibrosis progression other than TGF-β1 for its close involvement in other processes[10–12]. In addition, a panel of endogenous TGF-β1 antagonists, including Klotho, proteoglycan, BMP-7, lipoprotein (a) [Lp (a)] and retinoic acid, have been identified and exhibited antifibrotic capacities[13–17]. These findings bring insight into a promising therapeutic strategy against renal fibrosis by targeting the endogenous antifibrotic factor(s) rather than deletion TGF-β1.

The JNK-associated leucine zipper protein (JLP, also known as SPAG9) is a scaffold protein belonging to the family of JNK-interacting proteins (JIPs) with 5 members of JIP, JIP1, JIP2, JIP3, and JIP4. JIPs are scaffold proteins possessing pleiotropic functions. JIPs regulate MAPK signaling transduction by structurally and functionally coordinating kinases into a certain signaling modules[18]. JIPs are also essential in cell molecules trafficking and vesicle transportation[19–23]. Recently, JIPs has also been found involved in autophagy regulation[24,25]. Compared with other JIPs, JLP interacts with broader range of signaling molecules such as JNK, p38 MAPK, transcription factors Max / Myc, PLK1, G protein Gα13, dynein, kinesin, and surface protein Cdo[20,26–30], rendering it in the center of cellular functions regulation. The expression of JLP is in a manner of tissue specific and cell status dependent[19,20,27,31–35]. JLP is highly expressed in mouse tissues of brain, lung, spleen, testis, and ovary, but its expression is low or negative in heart, liver, epididymis, and uterus[32,33]. Analogously, JLP is positively expressed in various human cancer or malignant diseases to increase cancer cells abilities of survival, proliferation, migration, and invasion[36–45], and its overexpression is of great interest for diagnosis and prognosis[39,41,46–55]. These studies herald the emerging of JLP as not only a promising therapy target but also a potential biomarker in cancer disease. However, the expression pattern and the role of JLP in fibrotic disease remain elusive. In a genome-wide transcript-level analysis in samples from individuals with CKD, 2497 transcripts were identified with significant change in their expression values in the tubulointerstitial samples from CKD patients[56]. Among those altered transcripts, a striking reduction of Spag9 (Jlp) transcript was found in tubulointerstitial samples from those patients. In addition, our preliminary study indicated that JLP seem to be involved in the kidney fibrosis regulation[57]. This phenomenon led us speculate that JLP possess the capacities of tumorigenicity promotion, as well as fibrosis resistance, and its upregulation or loss contributes to the distinct different diseases of tumor or fibrosis.

We evaluated the role of JLP in the renal fibrosis regulation based on in vivo studies on unilateral ureteral obstruction (UUO) challenged mice with jlp global or TEC-specific deficiency, or with jlp TEC-specific transgenic overexpression and CKD patients undertaken renal biopsy, and in vitro study on cultured TECs. The results showed that JLP plays a crucial role in governing renal fibrosis through the mechanisms of negatively regulating TGF-β1 expression to counteract TGF-β1 initiated effects on ECM production, EMT, cell cycle arrest, apoptosis, as well as autophagy on TECs. These findings would help to deepen and broaden our perspective on the mechanism of renal fibrosis.

## Results

**Jlp deficiency exacerbates UUO induced renal fibrosis.** To investigate the role of JLP in renal fibrosis, we established the unilateral ureteral obstruction (UUO) mouse model in Jlp wild-type ($Jlp^{+/+}$) mice and Jlp deficient ($Jlp^{-/-}$) mice. We found that UUO led to increase the number of injured tubules and percentage of interstitial fibrosis in kidneys from $Jlp^{-/-}$ mice compared to kidneys from $Jlp^{+/+}$ mice as examined by hematoxylin-eosin (HE) staining and masson trichrome staining (MTS) (Fig. 1a). The fibrotic lesion was more severe in kidneys from $Jlp^{-/-}$ mice than that in kidneys from $Jlp^{+/+}$ mice characterized by increased the expression of the fibrotic molecular markers, including fibronectin (FN), collagen-I and a-SMA, in $Jlp^{-/-}$ kidneys compared to those in $Jlp^{+/+}$ kidneys as examined by Immunofluorescence (IF) staining (Fig. 1b), qPCR (Fig. 1c) and western blotting analysis (Fig. 1d and Supplementary Fig. 7A). In addition, although UUO could induce tubular cell apoptosis and macrophage infiltration in kidneys as examined by immunohistochemistry (IHC) staining for cleaved caspase-3, a marker of apoptosis, and by IF staining for F4/80, a marker of pan macrophage, respectively, however, UUO induced more severe tubular cell apoptosis and macrophage infiltration in kidneys from $Jlp^{-/-}$ mice than those in kidneys from $Jlp^{+/+}$ mice (Fig. 1e). These results suggested that depletion of JLP aggravated renal fibrosis in UUO mice.

$Jlp^{-/-}$ mice were constructed by knocking out Jlp gene globally, which resulted in Jlp deficiency in both renal intrinsic cells and renal extrinsic cells. To determine if loss of JLP in renal cells or external renal cells worsen renal fibrotic injury in UUO mice, $Jlp^{+/+}$ and $Jlp^{-/-}$ mice were irradiated with X-ray to deplete bone marrow cells and then transfused with cogenetic normal bone marrow, followed by UUO treatment (Supplementary Fig. 3). We found that $Jlp^{-/-}$ mice, although being replenished with JLP of normal bone marrow, still manifested with more severe kidney fibrotic lesion after UUO treatment, than that of $Jlp^{+/+}$ mice, which was assessed by means of MTS staining, the expression of α-SMA with IHC staining and western blotting assay with FN, collagen-I and a-SMA (Supplementary Fig. 3). These results suggested that loss of JLP in renal intrinsic cells was responsible for aggravated renal fibrosis in UUO mice.

**Conditional knockout of Jlp results in enhanced fibrosis.** To further investigate the role of JLP expressed by TECs in the kidney fibrosis, we established UUO mouse model in Jlp

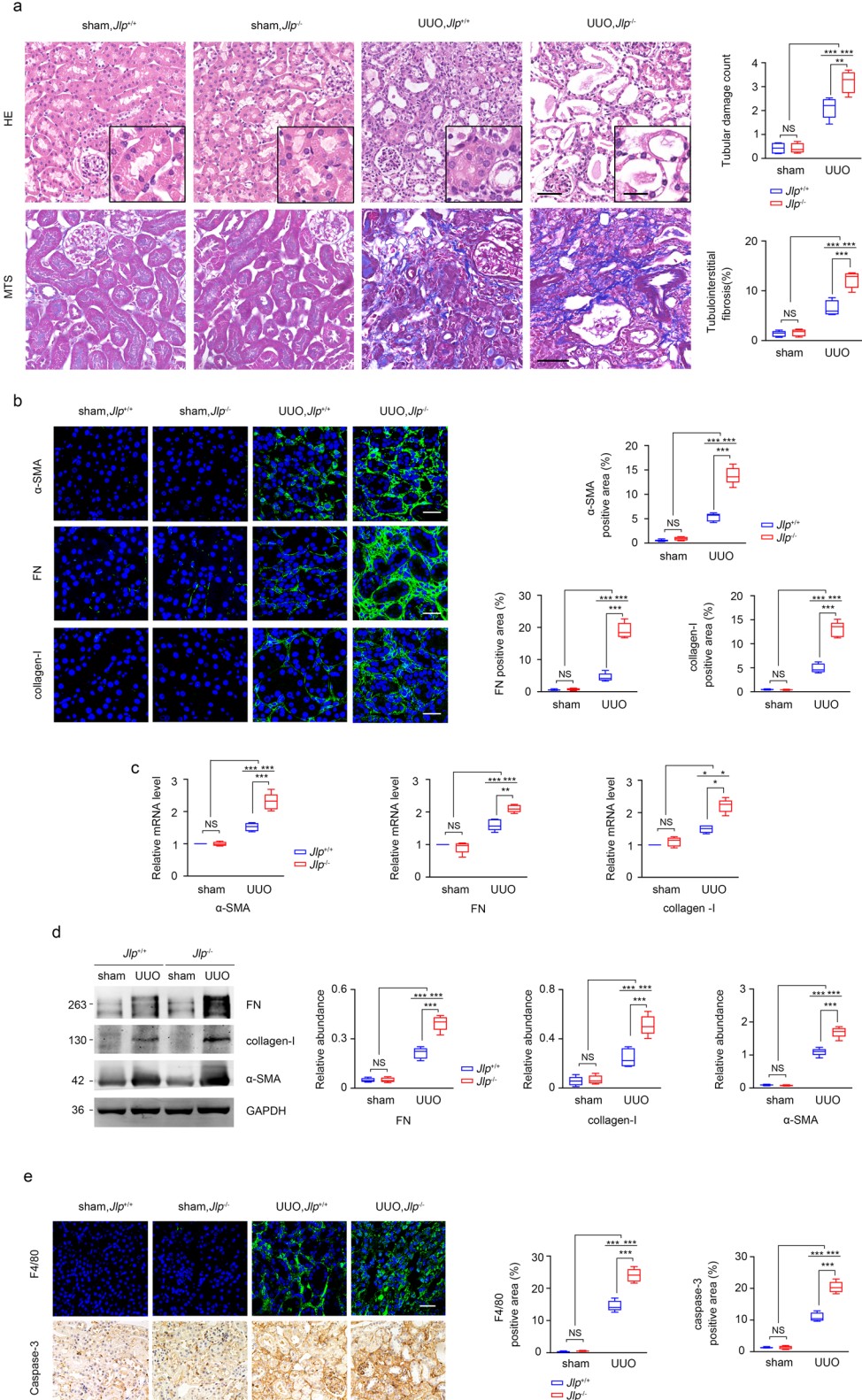

conditional knockout mice under the control of Ksp-Cre (*Jlp*flox/flox: Ksp-Cre mice, named as *Jlp*cKO), which express Cre recombinase specifically in TECs. The *Jlp*flox/flox mice without Cre expression (named as *Jlp*flox/flox) were used as control mice. Depletion of JLP exclusively in TECs resulted in more severe lesion of renal fibrosis in UUO kidneys of *Jlp*cKO mice than that in UUO kidneys of *Jlp*flox/flox mice as examined by HE and MTS

(Fig. 2a). In consistent, the expression of fibrotic related molecules, including FN, collagen-I and a-SMA, were increased in the UUO challenged kidneys from *Jlp*cKO mice than those in kidneys from *Jlp*flox/flox mice as measured by IHC staining, IF staining, and western blotting, respectively (Fig. 2b–d and Supplementary Fig. 7). Moreover, the TGF-β expression, as measured by IHC, western blotting and qPCR, was strongly increased in *Jlp*cKO mice

**Fig. 1 *Jlp* global deficiency aggravated UUO-induced kidney fibrosis. a** Representative images (five visual fields for each tissue analyzed) of HE and MTS of renal tissue section from indicated groups (left panel) and quantification of tubular lesion and interstitial fibrosis (right panel). Scale bar, 50 μm (insets, 10 μm). $n = 5$. **b** Representative images (five visual fields for each tissue analyzed) of IF staining for α-SMA and FN (green), and IHC staining for collagen-I in the indicated renal tissue sections. Cell nuclei are visualized by co-staining with DAPI. Scale bar, 50 μm.The positive areas of indicated protein were further presented in quantification (Right panel). $n = 5$. **c** *α-sma*, *Fn*, and *Collagen-I* mRNA level in the indicated kidney samples were measured by qPCR and normalized by *Gapdh* mRNA level. Expression of relative amounts of genes was calculated by the comparative CT method (2-△△CT) with the *Jlp*[+/+] sham group normalized to a fold value of 1. $n = 5$. **d** Western blotting analyses the expression of indicated proteins in the indicated kidney samples. GAPDH was set as loading control. The indicated band intensity of western blotting was normalized to the relevant band intensity of GAPDH. $n = 5$. **e** Representative images (five visual fields for each tissue analyzed) of IF staining for F4/80 (green), and IHC staining for Caspase-3 in the indicated kidney samples (left panel) and quantification of F4/80 and Caspase-3 expression based on IF or IHC staining. Scale bars, 50 μm. $n = 5$. Quantitative data are expressed as the mean ± s.e.m. Two-way ANOVA was applied for two-group comparisons. NS = no significant difference, *$P < 0.05$, **$P < 0.01$, ***$P < 0.001$.

compared with control kidneys (Fig. 2e–g and Supplementary Fig. 7b). These in vivo results from both the conventional and conditional knockout of *Jlp* in mice strongly suggested that TECs expressed JLP plays a critical role in regulating renal fibrosis.

**JLP is downregulated in fibrotic kidney tissues**. To determine if and how JLP plays a role in renal fibrosis in wildtype mice under the UUO, firstly we examined the expression and distribution of JLP in kidney. We found that JLP was mainly expressed in renal tubules, whereas its expression is weak or negligible in other sections of both mouse and human kidneys, such as glomeruli and interstitium (Supplementary Fig. 4). We then examined the expression of JLP in kidneys from fibrotic mice. We found that JLP protein expression was downregulated in kidneys at day 7 after UUO, and more obviously decreased at day 14 after UUO compared to that in sham operation and age matched kidneys (Fig. 3a, b, e). Next, we found that the expression of JLP protein was decreased in kidneys from CKD patients at stage 3 and 5 compared to that in normal human kidneys. The expression of JLP was reduced more obviously in kidneys from stage 5 CKD patients (Fig. 3c, d and Supplementary Fig. 7c). Furthermore, the *Jlp* mRNA levels in UUO kidneys and in kidneys of advanced CKD patients were also decreased compared to the controls (Fig. 3f, g). Our results suggested that reduced JLP expression is associated with the development of renal fibrosis.

To investigate the potential mechanism for the downregulation of JLP during UUO induced renal fibrosis, we treated cultured HK-2 cells, in which JLP was abundantly expressed, with TNF-α, TGF-β1, and FGF-2, which are the key initiators of inflammatory and fibrotic lesions[58,59]. We found that treatment with TGF-β and FGF-2, but not TNF-α, negatively regulated JLP protein and mRNA expression in HK-2 cells (Fig. 3h–j, and Supplementary Fig. 7c), in which TGF-β regulated the expression of JLP in a dose dependent manner (Fig. 3h, j). these results indicate that fibrotic factors TGF-β1 and FGF-2, which are over-produced during renal fibrosis progression, are contribute to the downregulation of JLP in the progress of renal fibrosis.

**JLP counteracts TGF-β1 expression and signaling in TECs**. To further address the mechanism of how JLP protects against renal fibrosis, we investigated whether JLP affects TGF-β1 signaling pathway in UUO challenged kidneys. We found that UUO treatment could induce remarkable higher expression of TGF-β1 protein and mRNA (Fig. 4a–c), as well as the activation of Smad signaling as presented by increasing the phosphorylation of Smad2/3 in kidneys from *Jlp*[−/−] mice compared to those in UUO kidneys from wild type mice (Fig. 4b). Next, we assessed the regulatory role of JLP on TGF-β1 expression in cultured HK-2 cells. As Fig. 4d, e showed that in normal conditions, TGF-β1 protein and mRNA expression were at low level and comparable in Jlp siRNA and control siRNA transfected

HK-2 cells. FGF-2 stimulation induced TGF-β1 expression (both in protein and mRNA), which was significantly higher in HK-2 cells with Jlp siRNA transfection than with control siRNA transfection. These results suggested that JLP is a negative regulator on TGF-β1 expression and TGF-β1 mediated Smad signaling activation.

**JLP counteracts TGF-β1 induced fibrotic responses in TECs**. We then investigated the role of JLP in TGF-β1 elicited fibrotic response in vitro. We found that even treatment of HK-2 cells with TGF-β1 induced considerable expression of fibrotic markers, including FN, collagen-I, and α-SMA, however, TGF-β1 treatment induced a more striking expression of these markers in *Jlp* knockdown HK-2 cells compared to those in the control siRNA transfected HK-2 cells as examined by western blotting, IF and qPCR (Fig. 5a–d). Due to that renal tubular cell cycle arrest and apoptosis are also key features of renal interstitial fibrosis, we therefore evaluated the effects of *Jlp* deficiency on cell cycle and apoptosis of HK-2 cells by flowcytometry. We found that TGF-β1 treatment induced significant G$_2$/M phase arrest and more cell apoptosis in *Jlp* knockdown cells (2.27-fold) than those in control siRNA transfected cells (Fig. 5e–h). Together, these results support a role of JLP in counteracting TGF-β1 induced fibrotic response, including ECM production, EMT, apoptosis, and cell cycle arrest in renal epithelial cells.

**JLP counteracts TGF-β1 induced autophagy in TECs**. It has been reported that UUO treatment induced autophagy in TECs[60], and persistent activation of autophagy in TECs promotes renal interstitial fibrosis during UUO[61–63]. To examine whether JLP protects against renal fibrosis through activation of autophagy in TECs, we examined the activity of autophagy in UUO-challenged kidneys from *Jlp* conditional knockout, *Jlp*[cKO] mice and the control *Jlp*[flox/flox] mice. We found that UUO treatment triggered the activation of autophagy in kidneys from both *Jlp*[cKO] and *Jlp*[flox/flox] mice as showed by increasing the expression of autophagy-related proteins, LC3-II and Beclin 1, and decreasing the expression of SQSTM1, a well-characterized substrate of autophagy, whereas more LC3-II and Beclin 1 were induced in UUO kidneys from *Jlp*[cKO] mice compared to the controls (Fig. 6a and Supplementary Fig. 7f). In addition, UUO treatment induced the activation of autophagy characterized by the LC3 pellet formation was mainly occurred in renal tubules as examined by LC3 IF staining (Fig. 6b) and by electron microscopy (Fig. 6c) to detect autophagic vacuoles in TECs. Compared with *Jlp*[flox/flox] mice, *Jlp*[cKO] mice exhibited increased levels of LC3 pellet formation in the TECs in response to UUO challenge. These data suggested that depletion of JLP in TECs augmented autophagy activity in UUO insulted kidneys.

Next, to further investigate the regulatory role of JLP on the activation of autophagy in TECs, we knocked down *Jlp* in HK-2

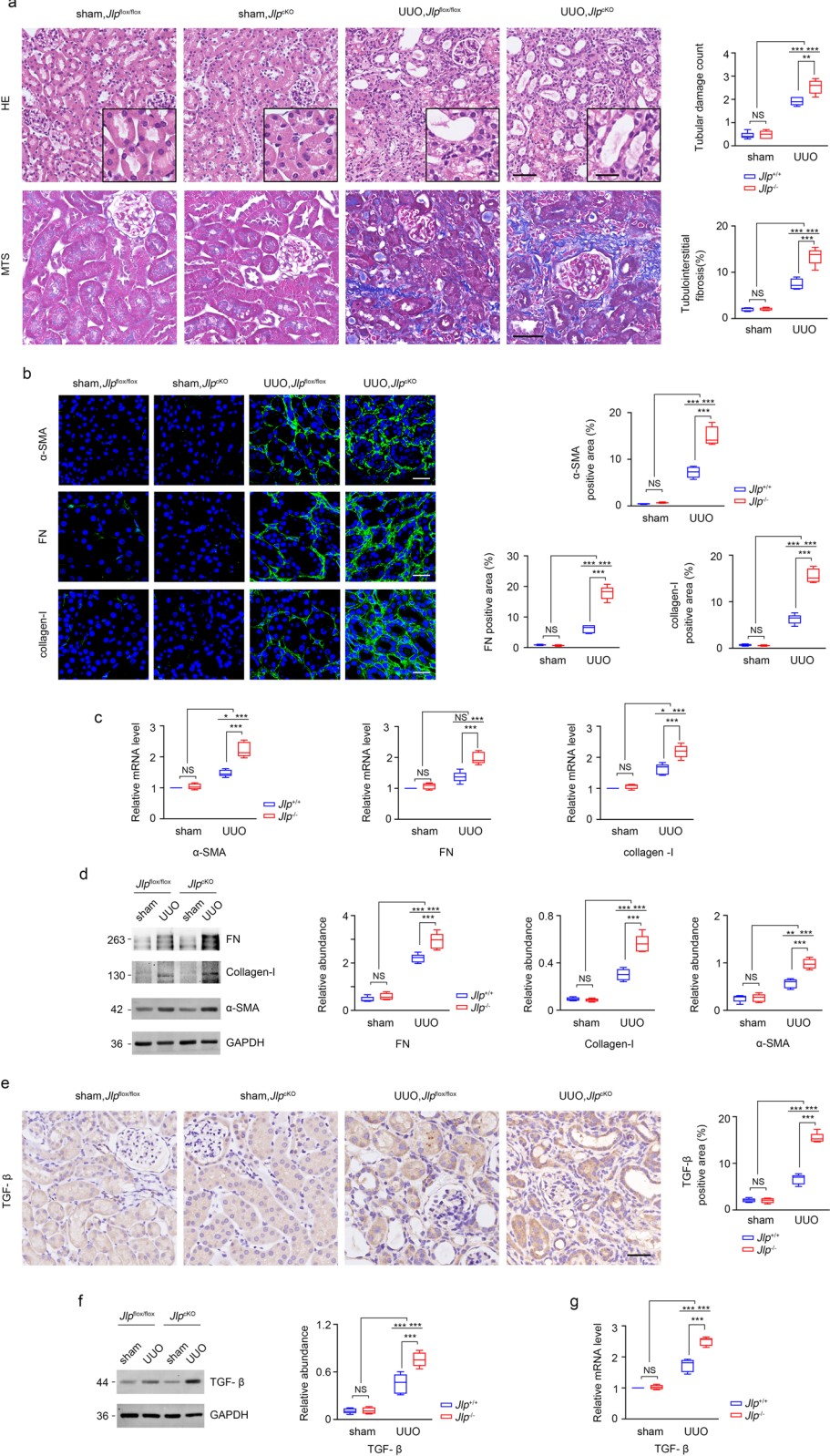

cells. Since TGF-β1 is an inducer of autophagy in TECs both in vitro and in vivo[61,64,65], Baf-a1 is an inhibitor of autolysosomal degradation pathway, and starvation can trigger autophagy in cells, the *Jlp* knockdown HK-2 cells and the control siRNA transfected HK-2 cells were further subjected to the treatments of TGF-β1, starvation, and Baf-a1, respectively. Treatment with TGF-β1

resulted in the increase of LC3-II and Beclin1, but slightly decrease of SQSTM1 in the *Jlp* knockdown HK-2 cells compared to those in HK-2 cells transfected with control siRNA and the *Jlp* knockdown HK-2 cells without TGF-β1 treatment (Fig. 7a), indicating that autophagic flux is increased with TGF-β1 treatment in the absence of JLP. Next, we found that both LC3-II upregulation and SQSTM1

**Fig. 2 TECs-specific deletion of JLP worsened the lesion of kidney fibrosis in UUO mice model. a** Representative images (five visual fields for each tissue analyzed) of HE and MTS of renal tissue from indicated groups (left panel). The tubular lesion and interstitial fibrosis were further presented in quantification (Right panel). Scale bars, 50 μm (inset, 10 μm). $n = 5$. **b** Representative images of IF staining for indicated proteins (green) in the indicated renal tissue. Cell nuclei are visualized by co-staining with DAPI. Scale bars, 50 μm.The positive areas of indicated protein were further presented in quantification (Right panel). $n = 5$. **c** α-sma, Fn, and Collagen-I mRNA level in the indicated kidney samples were detected by qPCR and normalized by Gapdh mRNA level. $n = 5$. **d** Western blotting analyzing the expression of indicated protein in the indicated kidney samples (Left panel), and the relative abundance of the indicated protein expression was normalized by GAPDH (Right panel). $n = 5$. **e** Representative images (five visual fields for each tissue analyzed) of IHC staining for TGF-β1 in the indicated renal tissue (Left panel) and quantitative data of the positive areas of TGF-β1 staining (Right panel). Scale bars, 100 μm. $n = 5$. **f** Western blotting analyzing TGF-β1 protein expression in the indicated kidney samples (Left panel) and quantification of the relative abundance of TGF-β1. $n = 5$. **g** Tgf-β1 mRNA level in the indicated kidney samples were detected by qPCR and normalized by Gapdh mRNA level. $n = 5$. Quantitative data are expressed as the mean ± s.e.m. Two-way ANOVA was applied for two-group comparisons. NS = no significant difference, $*P < 0.05$, $**P < 0.01$, $***P < 0.001$.

downregulation were comparable in JLP expressed and Jlp knockdown HK-2 cells treated with serum starvation, whereas serum starvation induced more Belin1 in Jlp knockdown HK-2 cells than that in control siRNA transfected HK-2 cells (Fig. 7b). In contrast to TGF-β1 treatment and starvation, treatment with Baf-a1 resulted in the upregulation of LC3-II, Beclin1, as well as SQSTM1 in Jlp knockdown HK-2 cells compared to those in control siRNA transfected HK-2 cells (Fig. 7c and Supplementary Fig. 7g). In addition, we found that TGF-β1 treatment induced more LC3 pellet formation in the Jlp knockdown HK-2 cells compared to that in HK-2 cells transfected with control siRNA as visualized by IF staining (Fig. 7d). Our results demonstrated that JLP is a negative regulator of autophagy in HK-2 cells, and JLP governs the autophagy activity partially through controlling the expression of Beclin1.

**JLP overexpression in TECs curbs the profibrotic effects**. To explore whether overexpression of JLP in TECs can rescue UUO induced renal fibrosis in vivo, we generated the Jlp transgenic mice with overexpression of JLP specifically in TECs in vivo (named as Cre+JlpTG). The normal level of JLP expression (Cre−JlpTG) was used as control. Both Cre+JlpTG and Cre−JlpTG mice were subjected to UUO-operation or sham-operation. JLP protein expression in kidney tissue was detected by western blotting, and renal fibrosis was assessed with histopathological manifestation and the expression levels of fibrotic related molecules. We found that although UUO treatment resulted in the downregulation of JLP protein in kidneys from both Cre−JlpTG and Cre+JlpTG mice, the expression levels of JLP were still more available in kidneys from Cre+JlpTG mice than those in kidneys from Cre−JlpTG mice (Fig. 8a). JLP overexpression in TECs significantly ameliorated the renal fibrosis as illustrated by HE and MTS (Fig. 8b). Accordingly, the protein and mRNA expression of fibrotic related markers, including FN, collagen-I and α-SMA, were remarkably reduced in UUO challenged kidneys from Cre+JlpTG mice than those in UUO kidneys from Cre−JlpTG mice (Fig. 8c–e). In addition, the upregulation of TGF-β1 protein and mRNA in fibrotic kidneys were reversed by overexpression of JLP specifically in TECs (Fig. 8f–h). We further found that the expression of LC3-II and Beclin 1 was significantly decreased and the expression of SQSTM1 was drastically increased in UUO treated kidneys from Cre+JlpTG mice compared to Cre−JlpTG mice (Supplementary Fig. 5), which was consistent with the remarkable reduction of autophagic vacuoles in TECs in kidneys from Cre+JlpTG mice, indicating that overexpression of JLP in TECs could mitigate UUO induced autophagy.

## Discussion
In this study, we identified that JLP, a scaffolding protein, is a potential endogenous antifibrotic factor in the development of renal fibrosis through counteracting the profibrotic effects of

TGF-β1. TGF-β1 is the critical driver in renal fibrosis for its formidable and comprehensive capability to promote ECM production, EMT, apoptosis and cell cycle on TECs[66], which has been suggested as a promising therapeutic target. However, targeting TGF-β1 in CKD patients by TGF-β1 ablation did not yield exciting results in preventing/decreasing renal fibrosis, suggesting the complexity of fibrosis mediated by TGF-β1 and the necessity to identify potential antifibrotic factors and mechanisms[4]. We found that the progression of fibrotic kidney diseases, either in UUO mouse model or CKD patients, was associated with TGF-β1 overexpression and JLP downregulation in kidney tissue.

Recently, several anti-fibrotic factors with the capacity to counteract the activities of TGF-β1 with different mechanisms have been identified in kidneys. For example, Klotho, predominantly produced by TECs, possess the potentials against the development of renal fibrosis, which is partly mediated by blocking TGF-β1 signaling through the mechanism of directly binding to the type-II TGF-β1 receptor to inhibit the binding of TGF-β1 to its cell surface receptors[14,67]. BMP-7, another protein highly expressed in kidney, has the capacity of anti-fibrosis by counteracting TGF-β1 initiated fibrotic effects, reducing ECM formation by inactivating matrix-producing cells, and promoting EMT[68,69]. Our present findings proved not only that JLP is an endogenous antifibrotic factor within the TGF-β1 antagonist family, but also that JLP executes its antifibrotic effects with different mechanisms as Klotho and BMP-7 do in many aspects. First, as the expression of JLP can also be observed in the distal tubules. However, the expression was more abundance in the proximal tubule (Supplementary Fig. 4). In contrast, Klotho is a membrane-bound protein which is highly expressed in the distal tubules but is weakly expressed in the proximal tubules in wild-type kidneys[70], and BMP7 is highly expressed in distal tubules, collecting ducts and podocytes[71]. Second, genetic deletion of JLP had no effect on the lifespan and development of mice except on reduction the fertility of male mice[33], whereas loss of Klotho or BMP-7 resulted in shortened lifespan and the serious impairment of kidney development[72–74]. Third, our study demonstrated that JLP has broad-spectrum of anti-fibrotic roles, such as inhibiting ECM production, EMT, apoptosis, cell cycle arrest, TGF-β1 expression, and autophagy in TECs. The roles of JLP in negatively regulating TGF-β1-initiated autophagy and inhibiting TGF-β1 expression in TECs has never been reported on Klotho and BMP-7.

Our studies revealed an interesting reciprocal negative regulation between JLP and TGF-β1 (Fig. 3and Fig. 4). It is reasonable to speculate that JLP and TGF-β1 form an essential axis to direct renal fibrosis development. The defect expression of JLP in TECs is the critical event leading to the failure of TGF-β1 restriction and contributes to the accelerated fibrosis. Considering the fact that JLP is predominantly expressed by TECs, while TGF-β1 and FGF-2 are expressed in abundant cell types, such as macrophages, TECs, and fibroblasts, we presume that the inhibition of TGF-β1 expression in TECs by JLP is offset by the

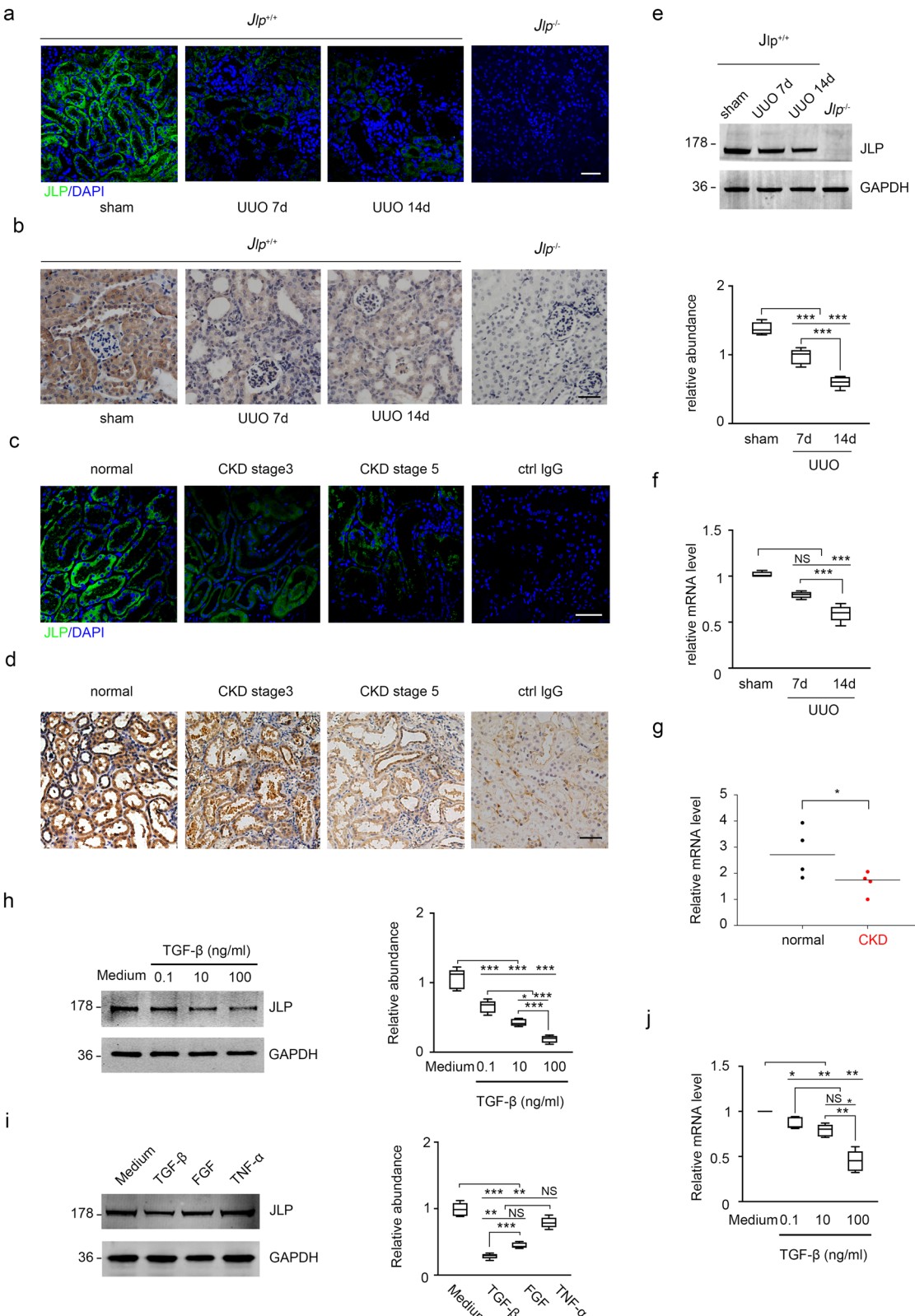

emerging of large amount of TGF-β1 from various origination, and then the prevailing production of TGF-β1 eventually downregulates the levels of JLP in TECs, resulting in the loss of cyto-protection by JLP. However, mice which Overexpression of JLP in TECs was more resistant to the UUO induced renal fibrotic lesion (Fig. 8), which provides a potential strategy of targeting JLP to prevent kidney fibrosis.

In the development of kidney fibrosis, interstitial inflammatory response to injuries is the early event and key trigger of inflammatory lesions. The persistent inflammation will be culminated with multifunctional inflammatory cells infiltration, myofibroblast formation, and growth factor and cytokines production[3,75]. Besides profibrotic cytokines, such as TGF-β1 and FGF-2, inflammatory cytokines TNF-α is also a critical factor in fibrogenesis[75]. To explore

**Fig. 3 Expression of scaffold protein JLP was decreased in fibrotic kidneys from the UUO model or CKD patients. a** Representative images (five visual fields for each tissue analyzed) of IF staining of JLP (green) in the renal cortex from indicated groups, $Jlp^{-/-}$ mice sets as negative control. Scale bars, 50 μm. $n = 5$ **b** Representative images (five visual fields for each tissue analyzed) of IHC staining of JLP in kidney from indicated group. $Jlp^{-/-}$ mice sets as negative control. Scale bars, 50 μm. $n = 5$ **c** Representative images (five visual fields for each tissue analyzed) of IF staining of JLP (green) in the renal cortex of kidney from healthy human biopsies and individuals with CKD. Ctrl IgG sets as negative control. Scale bars, 10 μm. **d** Representative images (five visual fields for each tissue analyzed) of IHC staining of JLP in the renal cortex of kidney from healthy human biopsies and individuals with CKD. Ctrl IgG sets as negative control. Scale bars, 10 μm. **e** Expression of JLP protein by western blotting of tissue lysates from kidney in indicated experimental groups and quantified, $n = 5$. **f** Relative expression of $Jlp$ gene from kidney in the indicated groups. Data are normalized to $Gapdh$ mRNA level. $n = 5$. **g** Relative $Jlp$ mRNA level was determined by qPCR in normal control kidney samples and kidney samples from individuals with CKD. $n = 4$. **h** Expression of JLP protein by western blotting of HK-2 cells lysates in indicated experimental groups (Left panel), and the relative abundance of JLP expression was quantified (Right panel). GAPDH sets as loading control. $n = 5$. **i** Expression of JLP protein by western blotting of HK-2 cells lysates in indicated experimental groups (Left panel), and the relative abundance of JLP expression was quantified (Right panel). GAPDH sets as loading control. **j** Relative $Jlp$ mRNA level was determined by qPCR in HK-2 cells from different groups as indicated. Data are normalized to $Gapdh$ mRNA level. $n = 5$. Quantitative data are expressed as the mean ± s.e.m. One-way ANOVA test with Bonferroni correction was used. NS = no significant difference, $*P < 0.05$.

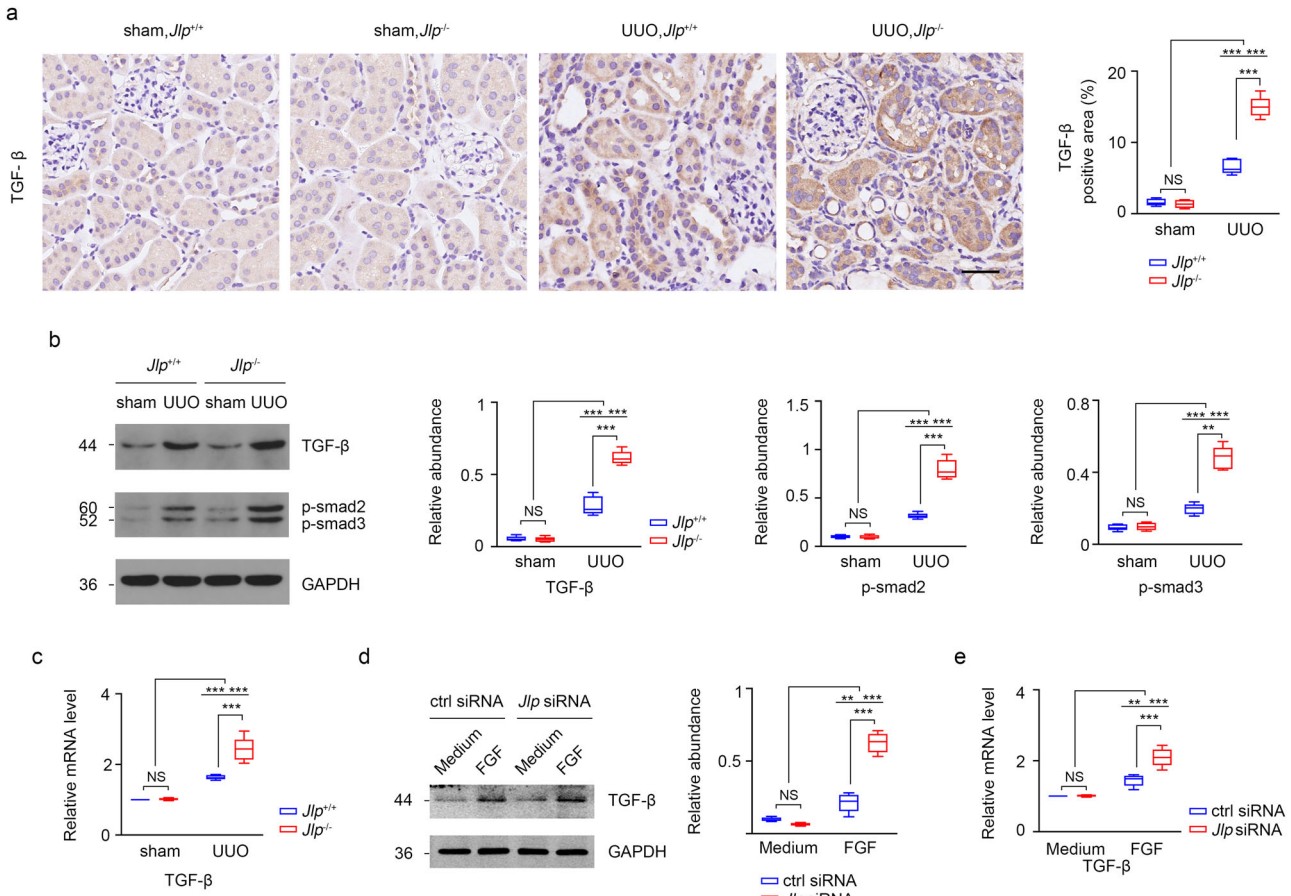

**Fig. 4 $Jlp$ deficiency resulted in enhanced TGF-β1 signaling activation in TECs. a** Representative images (five visual fields for each tissue analyzed) of IHC staining of TGF-β1 in kidneys from the indicated groups (left panel) and quantitative data of the positive areas of TGF-β1 staining (right panel). Scale bar, 100 μm. $n = 5$. **b** Western blotting analyzing the expression of TGF-β1, phospho-smad2 (p-Smad2) and phospho-smad3 (p-Smad3) in kidneys from indicated groups (left panel) and the quantification of band intensity (right panel, normalized by indicated GAPDH band). $n = 5$. **c** Relative $Tgf$-$β$ mRNA level (normalized by $Gapdh$ mRNA level) was determined by qPCR in kidneys from indicated groups. $n = 5$. **d, e** Jlp or control siRNA transfected HK-2 cells were stimulated with FGF-2 (10 ng/ml for 6 h) and subjected to western blotting and qPCR to assay the TGF-β1 protein and mRNA expression. Relative abundance of TGF-β protein in the western blotting (**d**) and relative $Tgf$-$β$ mRNA level (**e**) were calculated. $n = 5$. Data are presented as mean ± s.e.m. Two-way ANOVA was applied for two-group comparisons. NS = no significant difference, $*P < 0.05$, $**P < 0.01$, $***P < 0.001$.

whether JLP expression is also influenced by inflammation factors, we studied the JLP expression in cultured TECs in the response to TNF-α stimulation and found that, unlike profibrotic cytokines TGF-β1 and FGF-2, inflammatory cytokine TNF-α had no impact on JLP expression in TECs (Fig. 3). However, we still could not rule out the possibility that some other inflammatory factors/cytokines may regulate the expression of JLP. For instance, our previous study

has demonstrated that JLP in Dendritic cells (DCs) can be upregulated under CD40 signaling activation[20]. For the fundamental protective role of JLP in renal fibrosis, it is highly significant to identify factors and mechanism that can enhance JLP expression in fibrotic kidneys.

Autophagy is a fundamental cellular response to unfavorable conditions, such as nutrient starvation, energy deprivation, or

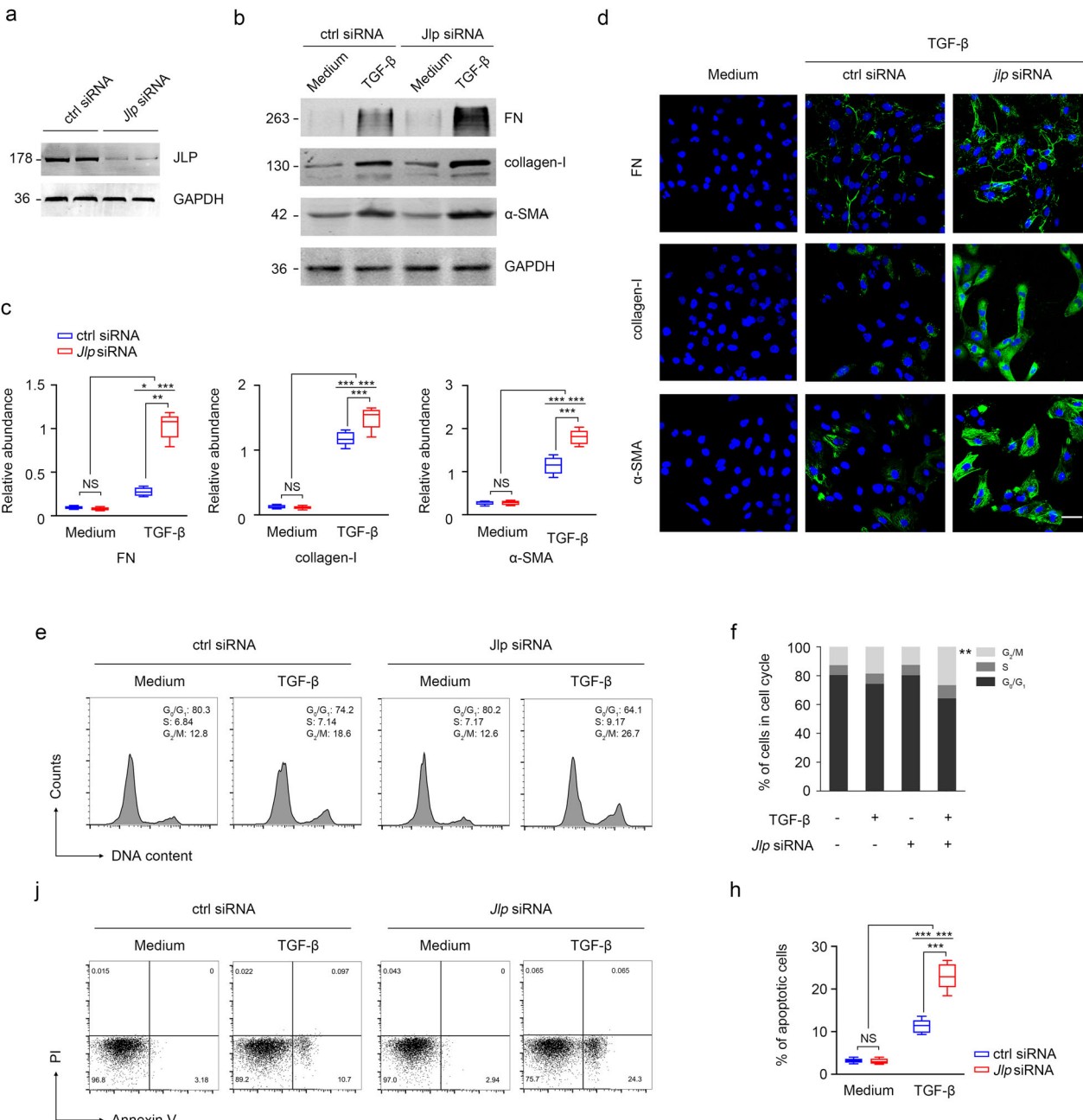

**Fig. 5 *Jlp* deficient HK-2 cells were more susceptible to TGF-β1 induced fibrotic responses in TECs.** Control or Jlp siRNA transfected HK-2 cells were treated with TGF-β (10 ng/ml) for 24 h and then subjected to western blotting, IF, or flowcytometric analysis to detect the expression of FN, collagen-I and α-SMA, or cell cycle stage and apoptosis. **a** *Jlp* was knockdown in HK-2 cells by Jlp siRNA transfection. $n = 3$. **b** Expression of FN, collagen-I, and α-SMA in HK-2 cells in different conditions were measured by western blotting analysis and relative band intensity of FN, collagen-I, and α-SMA were calculated in **c**. GAPDH sets as loading control. $n = 3$. **d** Representative images (five visual fields for each sample analyzed) of IF staining of FN, collagen-I, and α-SMA (green) in HK-2 cells under indicated conditions. Cell nuclei were visualized by co-staining with DAPI. Scale bars, 50 μm. $n = 3$. **e, f** Cell cycle measured by flowcytometric analysis in HK-2 cells under indicated conditions. Percentage of cells in different cell cycles were calculated by FlowJo, $n = 3$. **j–h** Cell apoptosis measured by flowcytometric analysis in HK-2 cells under indicated conditions. Percentage of apoptotic cells were calculated by FlowJo, $n = 3$. Data are presented as mean ± s.e.m. Two-way ANOVA was applied for two-group comparisons. NS = no significant difference, *$P < 0.05$, **$P < 0.01$, ***$P < 0.001$.

pathogen infection, which functions to maintain cell homeostasis by degrading cytoplasmic components via the formation of autophagosomes followed by autolysosomes[76]. This response is a primarily protective mechanism for cell survival, but it can also lead to cell death in case of autophagy being out of control[76,77]. Basal constitutive autophagy in proximal tubular cells appears to keep cell integrity[78]. In nephrotoxic and ischemic kidney injury

models, autophagy induced in proximal tubules plays a renal protective role[79–81]. However, persistent autophagy is able to promote renal fibrosis through triggering tubular atrophy, interstitial inflammation, and the production of profibrotic factor FGF-2[61,63,82–84]. TGF-β1 has been documented as inducer of autophagy on renal tubules in vitro or in vivo[61], which is also confirmed in this study. Most importantly, we present for the first

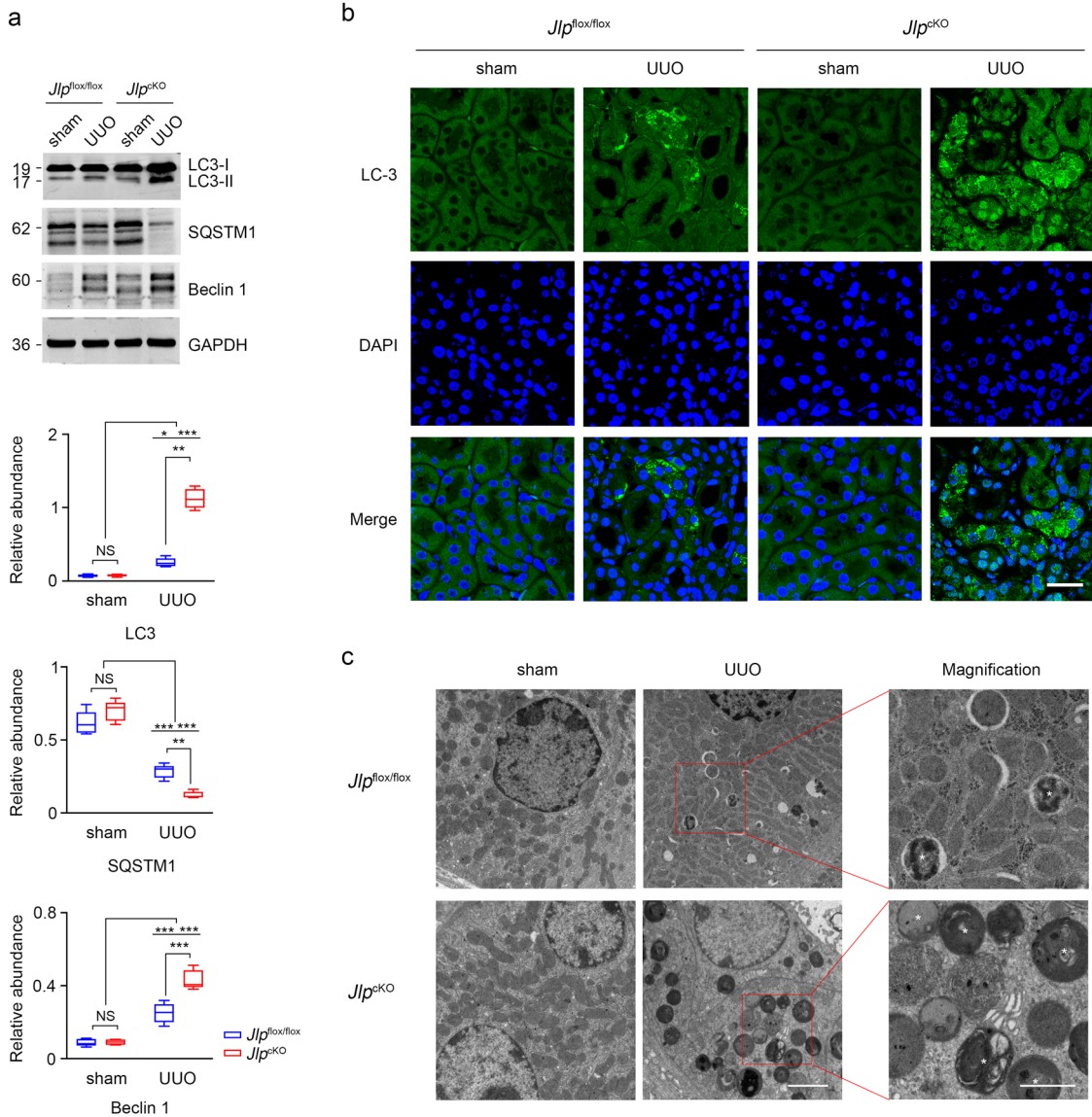

**Fig. 6 Deletion of scaffold protein JLP results in enhanced activation of autophagy in UUO challenged kidney. a** Expression of LC3 and Beclin 1 protein in kidney tissue lysates from indicated experimental groups were analyzed by western blotting (upper panel). Relative abundance of indicated protein expression was calculated (lower panel). GAPDH set as loading control. $n = 5$. **b** Representative images (five visual fields for each tissue analyzed) of IF staining of LC3 (green) in the renal cortex of kidney from indicated groups. Cell nuclei were visualized by co-staining with DAPI. Scale bar, 50 μm. $n = 5$. **c** Electron microscopic images of autophagic vacuoles in TECs from indicated groups. White asterisks indicate autophagic vacuoles. Scale bars, 2 μm. (insets, 1 μm). $n = 5$. Data are presented as mean ± s.e.m. Two-way ANOVA was applied for two-group comparisons. NS = no significant difference, *$P < 0.05$, **$P < 0.01$, ***$P < 0.001$.

time, to our knowledge, that JLP is a negative modulator of autophagy in TECs (Figs. 6 and 7). Our results suggest that JLP may be through its negative effect on TGF-β1 to regulate autophagy magnitude to protect TECs from death. However, JLP mediated renal-protection could be daunted by the negative feedback loop between the upregulation of TGF-β1 or FGF-2 and the downregulation of JLP, thereby leading to the uncontrolled TECs autophagy. Although the precise molecular mechanism of autophagy inhibition/induction by JLP on TECs remains unclear, our result that that *Jlp* deficiency resulted in higher level of Beclin 1 expression under various conditions both in vivo and in vitro (Figs. 6 and 7), suggesting that JLP might harness autophagy activity through regulating the expression of beclin-1.

JLP is a conserved multi-functional scaffolding protein, which constitutes a platform to tether different signaling molecules to regulate signaling transduction and vesicle trafficking[19,20,26,28,85],

and thus plays a key role in regulating various cellular processes and biological functions. The aberrant expression and dysfunction of JLP should impair its ability of maintaining cell homeostasis and contribute to the pathogenesis of human diseases. Recent studies found that overexpression of JLP is essential for tumor cells proliferation, migration, and invasion, and has been noticed as a hallmark in tumor cells or immortal cell lines, or in various human malignant diseases[36–45]. Directly targeting JLP achieved promising effects on suppressing tumor growth and metastasis[42,43,86]. Unlike tumors, organ fibrosis, which is always leading to various organ injury and failure, represents another human disease with features of parenchymal cell decay in vitality and decline in number. Whether JLP is involved in the regulation of organ fibrosis should be addressed. In a previous study of genome-wide transcriptional analysis on kidney tissues from CKD patients indicated that the expression of *Jlp* transcript was

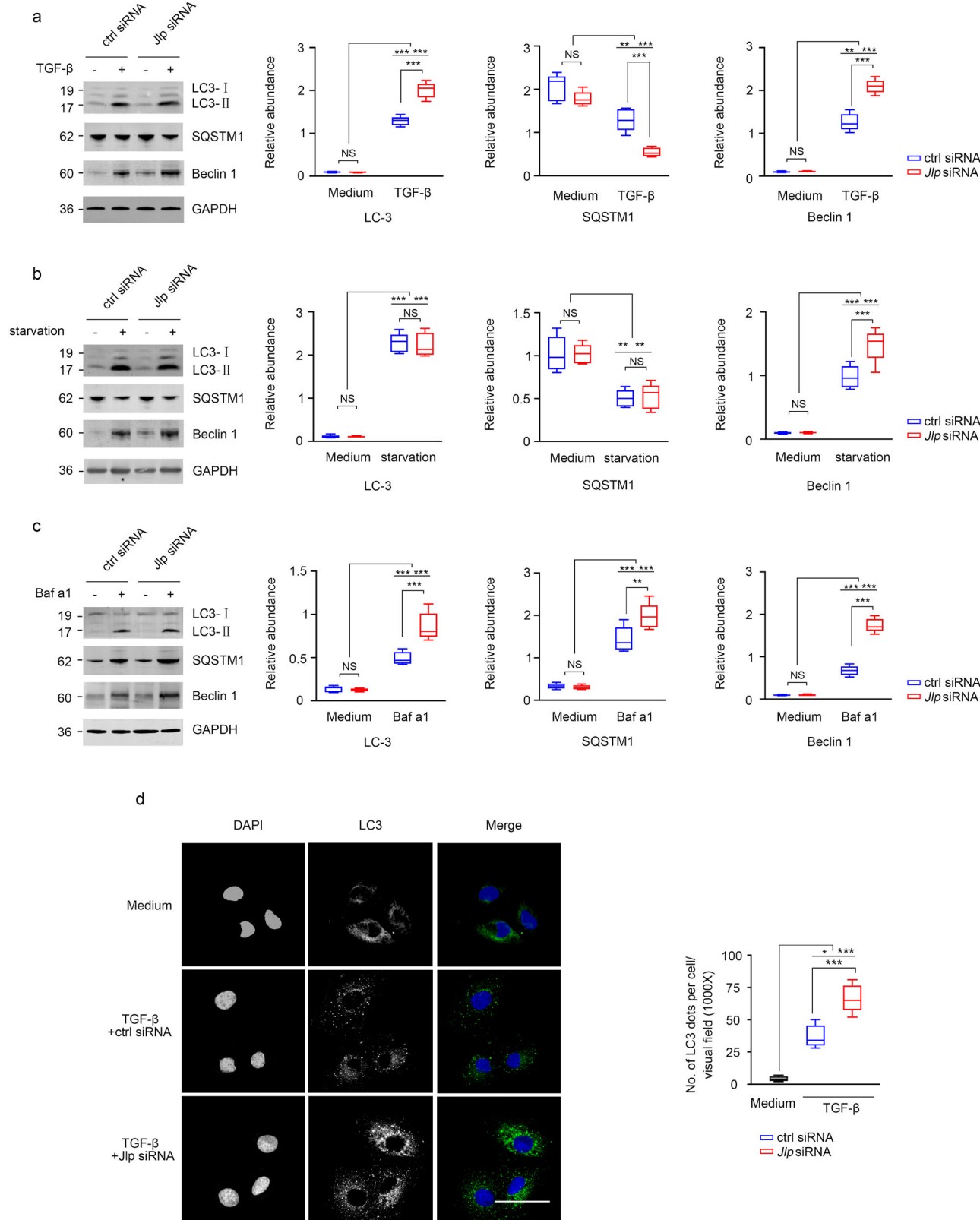

downregulated in tubulointerstitial samples from individuals with advanced CKD, suggesting that JLP may be a key player in CKD, a fibrotic disease. In this study, we found that JLP downregulation was associated with the renal fibrosis in both mouse UUO model and advanced CKD patients (Fig. 3), and JLP expression enable TECs resistant against TGF-β1 induced fibrotic effects on ECM

production, EMT, apoptosis, cell cycle arrest, as well as autophagy (schematic illustrated in Supplementary Fig. 6). This is the first study, to our knowledge, to show that JLP plays a beneficial role in fibrotic disease. Our findings, along with previous studies in tumor, extend our understanding of the positive and negative role of JLP in fibrosis and tumors, respectively.

**Fig. 7 JLP negatively regulates autophagy activation in TECs.** Control or Jlp siRNA transfected HK-2 cells were stimulated under indicated conditions and then subjected to western blotting or IF staining to check the autophagy activity. **a** HK-2 cells with indicated transfections were stimulated with or without TGF-β (10 ng/ml) for 24 h, and followed by western blotting to detect the expression of LC3, Beclin 1 and SQSTM1 protein. GAPDH sets as loading control. Relative abundance of indicated proteins was calculated (normalized by indicated GAPDH band). $n = 3$. **b** HK-2 cells with indicated transfections were starved for 24 h or not, and followed by western blotting to detect the expression of LC3, Beclin 1 and SQSTM1 protein. GAPDH sets as loading control. Relative abundance of indicated proteins was calculated (normalized by indicated GAPDH band). $n = 3$. **c** HK-2 cells with indicated transfections were stimulated with or without Baf a1 (10 nM) for 24 h, and followed by western blotting to detect the expression of LC3, Beclin 1 and SQSTM1 protein. GAPDH sets as loading control. Relative abundance of indicated proteins was calculated (normalized by indicated GAPDH band). $n = 3$. **d** HK-2 cells with indicated transfections were stimulated with or without TGF-β (10 ng/ml) for 24 h, and followed by IF to detect the formation of pelleted LC3, Cell nuclei was visualized by co-staining with DAPI (Left panel). Quantification of LC3 pellets per cells was presented (Right panel). $n = 3$. Scale bar, 50 μm. Data are presented as mean ± s.e.m. Two-way ANOVA was applied for two-group comparisons. NS = no significant difference, $*P < 0.05$, $**P < 0.01$, $***P < 0.001$.

## Methods

**Reagents**. Recombinant human TGF-β, recombinant FGF-2 and recombinant human TNF-α were purchased from Biolegend (San Diego, CA). Rapamycin was from ENZO life science (Loerrach, Germany). Bafilolycin A1 was from Invivogen. FITC-PHA was from Vector (FL-1111). FITC-PNA was from Sigma (L7381). DAPI was from Sigma (D9542). Alexa Flour 488, 594 or 647 conjugated anti-mouse or anti-rabbit IgG were from Jackson ImmunoResearch Laboratories (West Grove, PA). The primary antibodies against JLP (ab12331, 1:1000), Fsp-1 (ab197896, 1:200), α-SMA (ab124964, 1:1000), Fibronectin (ab45688, 1:500), Collagen-I (ab34710, 1:1000), Ki67 (ab16667, 1:250), TGF-β (ab92486, 1:500), phospho-Smad2 (ab188334, 1:1000), phospho-Smad3 (ab52903, 1:1000) and p62 (ab56416, 1:200) were all from Abcam (Cambridge, MA). Anti-LC3 antibody (ab51520, 1:100, Abcam) was used for immuno-staining and anti-LC3 antibody (L7543, 1:1000, Sigma) was used for western blotting, respectively. Anti-Nephrin and anti-F4/80 (123120, 1:100) antibodies were from Progen and Biolegend, respectively. Anti-Caspase-3 (#9664, 1:1000), anti-Beclin 1 (#3738,1:1000) was from Cell signaling Technology (Danvers, MA). Anti-GAPDH (sc-365062, 1:2500) was purchased from Santa Cruz (Santa Cruz, CA).

**Mice and animal models**. Jlp Wild type ($Jlp^{+/+}$) and Jlp global deficient ($Jlp^{-/-}$) mice were generated by inbreeding $Jlp^{+/-}$ mice as previously described[20]. $Jlp^{+/+}$ and $Jlp^{-/-}$ 8–10 weeks-of-age weight-matched and sex-matched littermates were used in the experiments. To generate TECs-specific Jlp deficient mice, we bred mice harboring a Ksp-Cre locus (Purchased from the Jackson Laboratory, California, USA), which express Cre recombinase specifically in TECs, with mice with LoxP-flanked alleles of $Jlp^{30}$ (hereafter as $Jlp^{flox/flox}$). The resulted mice with genotype of Ksp-Cre+/$Jlp^{loxP/loxP}$ were confirmed to be TEC-specific Jlp defective and hereafter referred as $Jlp^{cKO}$ mice (Supplementary Fig. 1).

In order to generate a TEC-specific Jlp transgenic (Tg) mice, the pDONR P2R-P3 plasmid, contained a Loxp-STOP-Loxp cassette and a CMV promoter, as well as mJlp cDNA domains, was linearized and microinjected into pronuclear-stage zygotes from female C57BL/6N mice, and transferred to pseudo-pregnant females. Tg founder mice containing mJlp gene were checked by PCR and maintain germline transmission. The established $mJlp$Tg mice (hereafter as $Cre^-Jlp^{TG}$) were then mated with Cdh16-Cre-ERT2 knock-in mice, a Cre mouse line which specifically express Cre-ERT2 in the presence of cadherin 16 promoter and were purchased from Cyagen Biosciences, Inc., Guangzhou, China. The resulted Cdh16-Cre-ERT2/$mJlp$Tg mice (hereafter as $Cre^+Jlp^{TG}$) were given 1.5 mg/day tamoxifen for 6 days (total 9 mg/mouse) at 4–5 weeks old mice by i.p. injection to induce JLP overexpressing with TEC-specific manner (Supplementary Fig. 2).

Mice were housed in specific pathogen-free conditions at the Center for Animal Experiments of Wuhan University. All animal experiments were approved by the Animal Ethics Review Board of Wuhan University and performed in accordance with the guidelines of the National Health and Medical Research Council of China.

For UUO model, mice were anesthetized with 30% isoflurane. The left ureter was exposed by a mid-abdominal incision. UUO was performed by tying off the left ureter with silk suture, sham-operated mice were undergoing same procedures without left ureter obstruction. The incision was closed and mice were kept breeding for 7 days or 14 days after surgery and sacrificed for further experiments.

In some conditions, $Jlp^{+/+}$ or $Jlp^{-/-}$ transgenic mice received irradiation of 6Gy X Ray or not using linear accelerator (VARIAN 23EX). The irradiated or non-irradiated mice were then subjected to operation of UUO and sacrificed at 7d later for kidney tissue collection as above described.

**Cell Culture and transfection**. Human TEC cell lines HK-2 was purchased from China Centre for Type Culture Collection and was maintained as previously described. Recombinant human TGF-β, recombinant FGF-2 and recombinant human TNF-α were purchased from Biolegend (San Diego, CA). HK-2 cells were treated with TGF-β or recombinant FGF-2 or recombinant human TNF-α for 24 h, respectively, and then were harvested and analyzed by qPCR and western blotting. Jlp siRNA and control siRNA was purchased from Qiagen (Germany). The siRNA transfection was carried out using HiPerFect transfection (Qiagen) according to customer's protocol. The efficiency of transfection was assessed by the protein

expression. The sequence used for knockdown of Jlp in this study was: 5′-CAGACCCGAGTGGAATCTTTA-3′. Bafilolycin A1 was from Invivogen. HK-2 cells treated with or without Bafilolycin A1, as well as the cells transfected with Jlp siRNA for 24 h,

**Western blotting**. We performed Western blotting on whole-cell or tissue lysates as described before[20]. Lysed samples were loading into 10% or 12% SDS gel for electrophrosis and transferred onto PVDF membrane (Merck Millipore) by X cell SureLock Mini-Cell system (Invitrogen). PVDF membranes contained proteins were then incubated with TBS containing 5% skim milk powder for 1 h at room temperature and incubated with indicated primary antibodies overnight at 4 °C. The next day membranes were incubated with IRDye 800CW secondary antibodies and scanned with Odyssey CLX Infrared Image System (all from LI-COR Biosciences, Lincoln, NE). Quantity One analysis software (Bio-Rad laboratories, Hercules, CA) were used to quantify the band intensity of some proteins, at least three bands from three individual experiments were calculated for band intensity analysis.

**Histology and Immunohistochemistry (IHC) examination**. Kidney samples were fixed in 4% paraformaldehyde (pH7.4) and then embedded in paraffin. For histology microscopy, tissue sections (4 μm) were stained with HE and MTS according to the manufacturer's protocols. The slides were examined by Olympus microscope (Olympus, Tokyo, Japan). Five visual fields (×200 magnification) of each group from five individual experiments were randomly selected. Hematoxylin-eosin staining was performed using standard procedures and renal tubules with the following histopathological changes were considered injured: loss of brush border, tubular dilation and disruption, cast formation and cell lysis. Tissue damage was examined in a blind manner and scored by the percentage of damaged tubules: 0, no damage; 1, <25%; 2, 25–50%; 3, 50–75%; 4, >75%. For Immunohistochemistry staining, sections were first de-paraffinized, hydrated and antigen retrieved then incubated with indicated primary antibodies. Five fields (×200 magnification) from individual groups were randomly selected and the percentage of positive stained area was quantitated using Image J ver. 1.37c analysis software (NIH, Bethesda, MD).

**Immunofluorescence (IF) staining**. For kidney tissues, paraffin-embedded kidney sections (4 μm) were de-paraffinized, blocking in PBS containing 10% normal donkey serum (Vector laboratories); immune-staining of cells were described before, briefly, the Cells cultured on coverslips were stimulated with indicated reagents, then wash with ice-cold PBS three times and fixed in 4% paraformaldehyde, and blocking in PBS containing 10% normal donkey serum. Samples were incubated with primary antibodies targeting JLP, FN, α-SMA, Collagen-I, LC3. The secondary antibodies were Alexa Flour 488, 594, or 647-conjugated. Samples were then staining with DAPI and Prolong Gold (Life technologies), the slides were visualized with confocal laser microscopy (Olympus FV1200, Tokyo, Japan). Five visual fields (x1000 magnification) of each group from three individual experiments were randomly selected, the fluorescence intensity of IF staining was quantified by Image J analysis software. For LC3 fluorescence dots quantification, five visual fields (×1000 magnification) of each group from three individual experiments were randomly selected and manually evaluated using Adobe photoshop.

**Transmission election microscopy**. Kidney samples were fixed with 4% paraformaldehyde and 1% glutaraldehyde. Tissues were cut into an ultrathin (40 nm) sections and stained with uranyl acetate. Sections were analyzed with a Hitachi H600 transmission electron microscopy (Hitachi, Tokyo, Japan). Autolysosome were marked as a single-membrane structure mixed with multi-vesicular bodies.

**Quantitativereal-time PCR (qPCR)**. Total RNA was extracted from kidney samples or cultured HK-2 cells using TRIzol reagents (Invitrogen). cDNA was synthesized with 1 μg total RNA. qPCR was performed by ABI7900 real-time PCR system (Illumina Eco, USA) using SYBR Green PCR Master Mix according to the

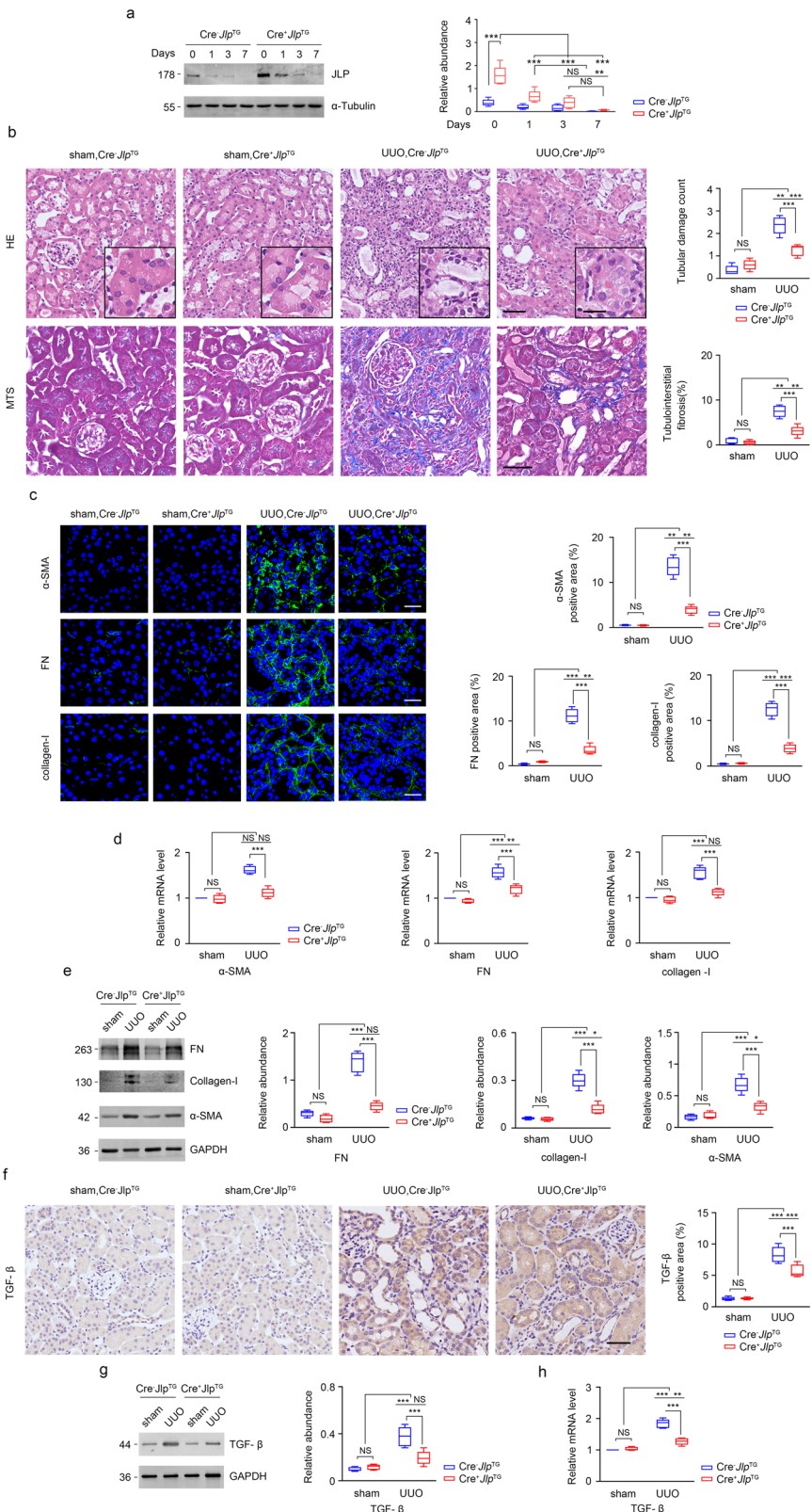

manufacturer's directions. Measurements of gene expression were standardized to the expression of the Gapdh housekeeping gene. Expression of relative amounts of genes was calculated by the comparative CT method (2-△△CT) with the control group normalized to a fold value of 1. The sequences of the primer pairs used in this study are shown in Supplementary Table 1.

**Flowcytometric analysis**. Cell apoptosis and cell cycle assay were performed using BD Annexin V apoptosis detection kit (556547) and BD Brdu flow kit

(552598) according to manufacturer's instructions, respectively. In brief, for cell apoptosis assay, cells were collected and wash with ice-cold PBS twice, cells then stained with FITC-annexin V and PI for 20 min in the dark then analyzed with flow cytometry. For cell cycle assay, cells were first co-cultured with Brdu (1 mM) for 30 min before indicated treatments, Cells then washed with ice-cold perm/wash buffer, fixed with cytofix for 15 min and re-suspended with cyto-perm/perm plus buffer for 10 min, Cells were stained with 7-AAD and anti-Brdu antibody following DNAase treatment. AccuriC6 (BD Biosciences) was used to

**Fig. 8 TECs-specific transgenic expression of *Jlp* ameliorated the lesion of UUO induced kidney fibrosis. a** Western blotting detecting JLP expression in renal tissue from indicated groups (left panel) and quantification of the relative abundance of JLP (Right panel). GAPDH sets as loading control. $n = 5$. **b** Representative images (five visual fields for each tissue analyzed) of HE and MTS of renal tissue from indicated groups (left panel). The tubular lesion and interstitial fibrosis were further presented in quantification (Right panel). $n = 5$. **c** Representative images (five visual fields for each tissue analyzed) of IF staining for indicated proteins (green) in the indicated renal tissue. Cell nuclei are visualized by co-staining with DAPI. The positive areas of indicated protein were further presented in quantification. $n = 5$. **d** *α-sma*, *Fn*, and *Collagen-I* mRNA level in the indicated kidney samples were detected by qPCR and normalized by *Gapdh* mRNA level. $n = 5$. **e** Western blotting analyzing the expression of indicated proteins in the indicated kidney samples (Left panel), and the relative abundance of the indicated protein expression was normalized by GAPDH (Right panel). $n = 5$. **f** Representative images (five visual fields for each tissue analyzed) of IHC staining for TGF-β1 in the indicated renal tissue (Left panel) and quantitative data of the positive areas of TGF-β1 staining (right panel). $n = 5$. **g** TGF-β1 protein expression by western blotting in the indicated kidney samples (Left panel) and quantification (Right panel). GAPDH sets as loading control. $n = 5$. **h** *Tgf-β1* mRNA level in the indicated kidney samples were detected by qPCR and normalized by *Gapdh* mRNA level. $n = 5$. Scale bar, 50 μm (inset, 10 μm). Data are presented as mean ± s.e.m. **a** One-way ANOVA test with Bonferroni correction was used. (**b**, **d**, **e**, **f**, **g**, **h**) Two-way ANOVA was applied for two-group comparisons. NS = no significant difference, $*P < 0.05$, $**P < 0.01$, $***P < 0.001$.

perform FACS and FlowJo software (Tree Star, Ashland, OR) were used to analyze data.

**Kidney biopsy specimens**. Human kidney biopsy specimens were obtained from patients with normal kidney function or under chronic impaired renal function (CKD patients), Human study was approved by clinical ethics committee of Renmin Hospital of Wuhan University (2012–44) and all patients were provided informed consent before inclusion in this study.

**Statistics and reproducibility**. No sample size calculation was conducted. Sample sizes for in vivo or in vitro experiments were set according to experiment experience, pilots and preliminary experiments, or referenced literature. Samples sizes of each experiment and numbers of each repeats are indicated in figure legends. In vivo experiments included contain ≥5 independent replicates. In vitro assays were performed at two or more replications. Data are expressed as mean ± s. e.m., SPSS ver. 17 (IBM SPSS, Chicago, IL) was used to analyze data. Ordinary one-way ANOVA multiple compare was used, Dunnett's test was used if more than two groups were compared. When comparing two variables, two-way ANOVA with Tukey's multiple-comparison test for two-group comparisons was applied. P-value < 0.05 was considered statistically significant.

**Reporting summary**. Further information on research design is available in the Nature Research Reporting Summary linked to this article.

## Data availability

The authors declare that the data supporting the findings of this study are available within the paper and its supplementary files or available from the corresponding author upon reasonable request. Source data are available in Supplementary Data 1.

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

## Acknowledgements

This work was supported by the grants from the National Natural Science Foundation of China (#81172793 and #81370800 to Dr. Huiming Wang, #81800603 to Dr. Qi Yan and #81800614 to Dr. Lu Zhang), the Key Project on Science and Technology Innovation of Hubei Province (#2019ACA137 to Dr. Huiming Wang). This study is also supported by National Institutes of Health (NIH) grant R01 DK084097 to Dr. Xiaogang Li.

## Author contributions

Q.Y., K.Z., Z.L., F.Q., and H.W. conceived and designed the research; Q.Y., K.Z., L.Z., Q.F., Z.C., S.L. B.D., and D.F. performed experiments; R.N., K.Y., Q.Y. and Z.C. constructed and maintained the genetically modified mice; Q.Y., K.Z., L.Z., Q.F., Z.C., S.L., and D.F. analyzed data; and Q.Y., G.D., X.L., and H.W. wrote the paper.

## Competing interests

The authors declare no competing interests.

## Additional information

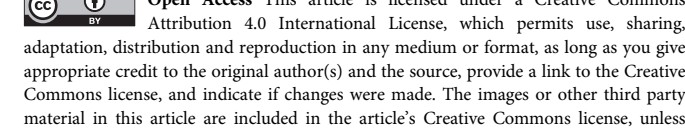

