## [Peer Review File · Communications Biology]

Reviewers' comments:

Reviewer #1 (Remarks to the Author):

Renal fibrosis is a common finding in CKD and strongly associated with functional prognosis. In this study, Yan, et al. revealed that JNK-associated leucine zipper protein (Jlp) negatively regulates profibrotic effects of TGF-b1 in renal tubular epithelial cells as an endogenous antifibrotic factor for the kidney. This novel molecule may have some clinical impacts on the nephrology field for the future, and seems of interest for the readers of Communications Biology. Prior to further consideration, however, the authors should appropriately respond to queries raised by this reviewer.

1. First of all, the authors should identify whether in vivo anti-fibrotic effects of Jlp depend on blockade of TGF-b1 signaling pathway or reduction in TGF-b1 expression. If the latter is true, application of therapeutic use of Jlp should be limited based on the lethal events in TGF-b1 knockout animals and the adverse side effects by TGF-b1 blockade treatments as the authors stated in the manuscript.
2. This reviewer feels that contribution of autophagy in tubular epithelial cells to renal fibrogenesis is still controversial, and remains to be clarified. Additionally, if Jlp is truly a negative modulator of autophagy in tubular epithelial cells, application of long-term, therapeutic use of Jlp might be dangerous because of enhancement of AKI susceptibility and forced aging.
3. In Figure 1e, almost all the kidney cells seem positive for cleaved caspase-3 in the UUO 7d model, irrespective of jlp gene expression. The background signals were too much high to be evaluated quantitatively.
4. In Figures 3j and 3l, to this reviewer, the band intensities of western blot of Fig.3j seem far different from the quantified data of Fig 3j.
5. In Figure 7b, the Beclin-1 band intensities of western blot also seem far different from the quantified data of Beclin-1 graph.
6. A number of typo-errors are found in the manuscript. The authors should carefully check the manuscript prior to submission.

Reviewer #2 (Remarks to the Author):

In the present manuscript, Yan and co-workers describe a novel role for the JNK-associated leucine zipper protein (Jlp) during kidney fibrosis. Specifically, by using multiple mouse models (Jlp full body KO, TEC-specific cKO and Jlp Tg) as well as in vitro model to silencing Jlp in tubular epithelial cells they showed that Jlp expression is downregulated in tubular epithelial cells during UUO-induced fibrosis and this correlates with a more severe fibrotic outcome. These findings were corroborated by the analysis of Jlp expression in kidney biopsies from CKD patients. Authors demonstrated that the mechanism underlying the protective role of Jlp during fibrosis is based on the existence of a negative feedback loop between TGFb1 and Jlp. Moreover they found that Jlp negatively regulates autophagy activation in TECs. The manuscript is interesting and the protective role of Jlp in kidney fibrosis is a novel finding. It however presents several weaknesses, as shown in my comments below, that should be addressed to strengthen the findings highlighted in this manuscript.

- The entire manuscript is based on using an extremely low number of mice. In fact, as indicated in the figure legends, n=3 mice per strain and experimental condition were analyzed. Considering that some of the phenotypic differences between WT, KO, cKO and Tg mice were mild, I highly doubt a size of 3 would ensure enough statistical power to draw the conclusions highlighted in this manuscript. Did the author truly perform experiments only on 3 mice per group? If so, I highly recommend to provide calculation of the statistical power and to increase the size accordingly.
- All data are presented as mean +/- SD. For certain type of analysis, such as the histological one, the fact that data were presented with the standard deviation is somehow concerning. Does that indicate that multiple analysis per each mouse have not been conducted? In fact, if multiple fields per view were analyzed and quantified for each mouse, they would be averaged together to represent the value for that mouse. Then, the average of the values of all the mice belonging to the same group would need to be averaged and that represents the final mean +/- the standard error of the mean (S.E.M.), instead of S.D. Please clarify this.
- Authors showed that loss of Jlp exacerbates TGFb-induced EMT and G2/M cell cycle arrest in vitro. Does the ameliorated phenotype observed when Jlp is restored result also from counteracting EMT and cell cycle arrest? Authors should provide evidences both in vivo (in the Jlp-Tg mice) and in vitro (by overexpressing Jlp in the HK2 cells).
- In vivo amelioration of autophagy activation in the Jlp-Tg mice. Authors should reinforce the result by performing LC-3 IF (as shown in Fig.6b) and presenting this data in the main figure panel to further support the major Jlp-dependent phenotype.
- Figures 1c, 2b, 5d, 8c: please add quantification of the aSMA, Fibronectin and Collagen I staining.

Minor points

- Both t-test and one/two-way ANOVA have been used for statistical analysis, however is not clear when one or the other was specifically used. Please specify in each figure legend what test has been used.
- Figure 1a figure legend is missing indication of the number of mice.
- In the figure legend relative to Figures 1d, 4d, 5b, 6a authors stated that "Number 1 and 2 indicate different mice in one group", however the figure panel does not show any multiplicity in the number of samples/per group shown. Instead, Figure 3e, where 1 and 2 are shown, does not contain that statement in the figure legend. Please review this.
- Figure legend relative to Figures 3a and 3c stated that images are from IHC staining, instead they are from IF staining. Please fix this.
- In the methods, authors indicated reference 27 as a support reference for the generation of the Jlp KO mice. However, Ref.27 is a review article, therefore most likely it is not the appropriate reference. Please review.

Reviewer #3 (Remarks to the Author):

Title: Renal tubular epithelial cells expression of JNK-associated leucine zipper protein (Jlp) is essential in counteracting TGF-beta1 initiated fibrotic effects

Reviewer's Comments:

The aim of this study was to evaluate the role of Jlp in fibrosis. The study is well planned and executed properly. However, there are certain issues with the manuscript.

Major comments:

1. The manuscript has not been arranged properly. The sequence of headings generates some confusion which should be avoided. The authors are advised to properly arrange the manuscript for

better understanding.

2. The referencing style is poor. There are many improper citations here and there. Please go through the whole manuscript and rectify it.

3. RT PCR data show relative mRNA levels of fibrotic markers as shown in the figure 1b. There should be a clarification regarding the data presentation if the data is relative to sham? How was the quantification done?

4. The data presentation is confusing in the manuscript. For example, in figure 2, the "*" denotes statistically significant difference between jlpflox and jlpckO or with sham. The authors should clearly mention the comparison of different groups in the figure legends to make it understandable. Without this comparison, it is difficult to understand whether there was a significant change after global or specific deletion of jlp in TECs.

5. There is too much of non-uniformity in the manuscript. Authors have used different styles for writing certain terms. JLP vs jlp, GAPDH vs gapdh, α -SMA vs α -sma, jlpflox/flox vs jlpflox/flox, fibronectin vs FN, H&E vs HE.

6. There is no quantification of stainings in figure 3a-3d. further, why the jlp KO group was not included in WB and other assays? It would have been interesting to show the relative expression of jlp after KO and UUO.

7. Throughout the manuscript, "abundance" has been written as "abandance".

8. The sequence of results needs to be rearranged as the present sequence creates some confusion while reading. The subheading 3 can be made subheading 1. The authors are also advised to show the in vitro autophagy induction results first, followed by in vivo results to make it clearer.

9. The quantification of stainings, IF, IHC should be more descriptive. What was the number of samples used? How many visual fields were considered? How the results were analysed?

10. Discussion part needs slight revision. The authors are advised to correlate their findings with earlier published reports.

Minor comments: The manuscript is full of typographical and grammatical errors E.g.

11. "in particularly, in fibroblasts and myofibroblasts multiple". Page 3

12. "examine the expression". Page 8

13. expressed 26, with TNF- α , TGF- β 1 and FGF-2, which are the key initiators of

14. inflammatory and fibrotic lesions 3,72. Last paragraph, page 8

15. "of which are over-produced" page 9

16. The authors need to modify the sentence "whereas more Jlp was identified in the proximal tubules in wildtype and normal kidneys". Page 15. As the expression of jlp can also be observed in the distal tubules. However, the expression was more in the proximal tubule. The discussion needs to be modified where the authors are comparing the jlp with klotho and BMP-7.

17. EMT production or ECM production? Page 18

18. Where was the chloroquine used? Was it used as standard autophagy inhibitor for comparison?

19. The human kidney specimen collection should be more descriptive or mention it is provided in supplementary data.

20. Primer or primary pairs? Page 24, rtPCR

21. Statistical analysis should be more descriptive in the figure legends. How the groups were compared?

22. Phosphor-smad3. Page 43

23. Why was the staining for F4/80 used? How to correlate it with the findings of present study?

24. Data are presented as mean \pm SDs. Page 43. It should be data.

25. Number 1 and 2 indicate each individual sample in each given group. There are no number 1 and 2. Page 45. Similarly, on page 47

Response to referees` comments

Reviewer #1 (Remarks to the Author):

Renal fibrosis is a common finding in CKD and strongly associated with functional prognosis. In this study, Yan, et al. revealed that JNK-associated leucine zipper protein (Jlp) negatively regulates profibrotic effects of TGF-b1 in renal tubular epithelial cells as an endogenous antifibrotic factor for the kidney. This novel molecule may have some clinical impacts on the nephrology field for the future, and seems of interest for the readers of Communications Biology. Prior to further consideration, however, the authors should appropriately respond to queries raised by this reviewer.

Response

Thank you very much for your in-depth review and the overall positive comments. We would address your concerning as follow.

1. First of all, the authors should identify whether in vivo anti-fibrotic effects of Jlp depend on blockade of TGF-b1 signaling pathway or reduction in TGF-b1 expression. If the latter is true, application of therapeutic use of Jlp should be limited based on the lethal events in TGF-b1 knockout animals and the adverse side effects by TGF-b1 blockade treatments as the authors stated in the manuscript.

Response

We had observed in vivo that when JLP was absent, the expression of TGF-b1 in the kidney of UUO mice was down regulated, and the activation of its signal pathway was also weakened. When JLP was overexpressed, TGF-b1 was negatively affected. Therefore, in vivo experiments are sufficient to prove the negative regulatory effect of JLP on TGF-b1. However, our study implied that JLP and TGF-b1 are mutually counteracted, and besides TGF-b1, JLP may be regulated by other molecules such as FGF-2. Therefore, we postulate that gaining expression of Jlp does not mean the thoroughly blocking of TGF-b1, but just ameliorating the over activation of TGF-b1 in renal fibrosis to some extent. We think restoring JLP function might be a more promising therapeutic strategy than directly targeting TGF-b1, and will not lead to lethal effect.

2. This reviewer feels that contribution of autophagy in tubular epithelial cells to renal fibrogenesis is still controversial, and remains to be clarified. Additionally, if Jlp is truly a negative modulator of autophagy in tubular epithelial cells, application of long-term, therapeutic use of Jlp might be dangerous because of enhancement of AKI susceptibility and forced aging.

Response

The role of autophagy in TECs in fibrogenesis is indeed a topic of debate, but accumulating evidences suggest that in nephrotoxic and ischemic kidney injury models, autophagy induced in proximal tubules plays a renal protective role, but persistent autophagy is able to promote renal fibrosis. In our opinion, autophagy is governed by series factors with promoting and antagonizing properties, among them Jlp and TGF-b1 are the representatives of a complex network system. Jlp belong to families that repress autophagy activity in TECs, the concomitant occurrence of rampant autophagy, worsened kidney fibrosis, and Jlp defects provide the circumstantial evidence of autophagy in TECs is detrimental in the process of renal fibrosis. Regarding the AKI

therapy, actually we had performed some experiments to observe the role of Jlp in those conditions, but found that Jlp in TECs displayed no change, this may indicate that no need of applying long-term, therapeutic use of Jlp in disease of AKI.

3. In Figure 1e, almost all the kidney cells seem positive for cleaved caspase-3 in the UUO 7d model, irrespective of jlp gene expression. The background signals were too much high to be evaluated quantitatively.

Response

Agree, the figure is not perfect to be on display. We had replaced it with the better one.

4. In Figures 3j and 3l, to this reviewer, the band intensities of western blot of Fig.3j seem far different from the quantified data of Fig 3j.

Response

We had checked our raw data and repeated the data collection and quantification, the modified figure is now presented with this manuscript revision.

5. In Figure 7b, the Beclin-1 band intensities of western blot also seem far different from the quantified data of Beclin-1 graph.

Response

We had checked our raw data and repeated the data collection and quantification, the modified figure is now presented with this manuscript revision.

6. A number of typo-errors are found in the manuscript. The authors should carefully check the manuscript prior to submission.

Response

We are sorry for the type-errors appeared in the manuscript, and had made the double check and correction in the current version.

Reviewer #2 (Remarks to the Author):

In the present manuscript, Yan and co-workers describe a novel role for the JNK-associated leucine zipper protein (Jlp) during kidney fibrosis. Specifically, by using multiple mouse models (Jlp full body KO, TEC-specific cKO and Jlp Tg) as well as in vitro model to silencing Jlp in tubular epithelial cells they showed that Jlp expression is downregulated in tubular epithelial cells during UUU-induced fibrosis and this correlates with a more severe fibrotic outcome. These findings were corroborated by the analysis of Jlp expression in kidney biopsies from CKD patients. Authors demonstrated that the mechanism underlying the protective role of Jlp during fibrosis is based on the existence of a negative feedback loop between TGF β 1 and Jlp. Moreover they found that Jlp negatively regulates autophagy activation in TECs. The manuscript is interesting and the protective role of Jlp in kidney fibrosis is a novel finding. It however presents several weaknesses, as shown in my comments below, that should be addressed to strengthen the findings highlighted in this manuscript.

Response

Thank you very much for your in-depth review and the overall positive comments. We would address your concerning as follow.

- The entire manuscript is based on using an extremely low number of mice. In fact, as indicated in the figure legends, n=3 mice per strain and experimental condition were analyzed. Considering that some of the phenotypic differences between WT, KO, cKO and Tg mice were mild, I highly doubt a size of 3 would ensure enough statistical power to draw the conclusions highlighted in this manuscript. Did the author truly perform experiments only on 3 mice per group? If so, I highly recommend to provide calculation of the statistical power and to increase the size accordingly.

Response

Thank you for your suggestions. According to your suggestion, we had performed additional experiments to increase the size to 5, which has been updated in the revised manuscript.

- All data are presented as mean \pm SD. For certain type of analysis, such as the histological one, the fact that data were presented with the standard deviation is somehow concerning. Does that indicate that multiple analysis per each mouse have not been conducted? In fact, if multiple fields per view were analyzed and quantified for each mouse, they would be averaged together to represent the value for that mouse. Then, the average of the values of all the mice belonging to the same group would need to be averaged and that represents the final mean \pm the standard error of the mean (S.E.M.), instead of S.D. Please clarify this.

Response

Agree and sorry for the mistake! We had performed the statistics analysis according to your suggestions and made correction in the revised manuscript. Thanks again!

- Authors showed that loss of Jlp exacerbates TGF β -induced EMT and G2/M cell cycle arrest in vitro. Does the ameliorated phenotype observed when Jlp is restored result also from counteracting EMT and cell cycle arrest? Authors should provide evidences both in vivo (in the Jlp-Tg mice) and in vitro (by overexpressing Jlp in the HK2 cells).

Response

Agree! But we are sorry currently could not present the entire data of EMT and cellcycle phenotype in the gaining expression of Jlp in vivo and vitro. For one reason is that these data will be organized in another paper, and second is that we now only got in vitro data. We would like to report that result soon.

- In vivo amelioration of autophagy activation in the Jlp-Tg mice. Authors should reinforce the result by performing LC-3 IF (as shown in Fig.6b) and presenting this data in the main figure panel to further support the major Jlp-dependent phenotype.

Response

Thank you for your valuable suggestion! But we are sorry currently could not present this figure because of some trouble we encountered in the experiments.

- Figures 1c, 2b, 5d, 8c: please add quantification of the aSMA, Fibronectin and Collagen I staining.

Response

We had reorganized the figures and add the quantification of those staining you required.

Minor points

- Both t-test and one/two-way ANOVA have been used for statistical analysis, however is not clear when one or the other was specifically used. Please specify in each figure legend what test has been used.

Response

As you requested we had made description in separate figure legend. Thank you pointed out the defects!

- Figure 1a figure legend is missing indication of the number of mice.

Response

It has been added.

- In the figure legend relative to Figures 1d, 4d, 5b, 6a authors stated that "Number 1 and 2 indicate different mice in one group", however the figure panel does not show any multiplicity in the number of samples/per group shown. Instead, Figure 3e, where 1 and 2 are shown, does not contain that statement in the figure legend. Please review this.

Response

The mistakes you pointed out have now been corrected. Thanks!

- Figure legend relative to Figures 3a and 3c stated that images are from IHC staining, instead they are from IF staining. Please fix this.

Response

Sorry for the mistake and thank you pointed it out! We had fixed it.

- In the methods, authors indicated reference 27 as a support reference for the generation of the Jlp KO mice. However, Ref.27 is a review article, therefore most likely it is not the appropriate

reference. Please review.

Response

Sorry for the mistake and thank you pointed it out! We had fixed it.

Reviewer #3 (Remarks to the Author):

Title: Renal tubular epithelial cells expression of JNK-associated leucine zipper protein (Jlp) is essential in counteracting TGF-beta1 initiated fibrotic effects

Reviewer's Comments:

The aim of this study was to evaluate the role of Jlp in fibrosis. The study is well planned and executed properly. However, there are certain issues with the manuscript.

Response

Thank you very much for your in-depth review and the overall positive comments. We would address your concerning as follow.

Major comments:

1. The manuscript has not been arranged properly. The sequence of headings generates some confusion which should be avoided. The authors are advised to properly arrange the manuscript for better understanding.

Response

Thank you for your advice. We had reorganized the manuscript.

2. The referencing style is poor. There are many improper citations here and there. Please go through the whole manuscript and rectify it.

Response

Sorry for the mistake and thank you pointed it out! We had fixed it.

3. RT PCR data show relative mRNA levels of fibrotic markers as shown in the figure 1b. There should be a clarification regarding the data presentation if the data is relative to sham? How was the quantification done?

Response

Agree and thanks! We had added the information in the figure legend.

4. The data presentation is confusing in the manuscript. For example, in figure 2, the “*” denotes statistically significant difference between jlpflox and jlpckO or with sham. The authors should clearly mention the comparison of different groups in the figure legends to make it understandable. Without this comparison, it is difficult to understand whether there was a significant change after global or specific deletion of jlp in TECs.

Response

Agree and thanks! We had improved the description to make the comparison clear in the figure legend.

5. There is too much of non-uniformity in the manuscript. Authors have used different styles for writing certain terms. JLP vs jlp, GAPDH vs gapdh, α -SMA vs α -sma, jlpflox/flox vs jlpflox/flox, fibronectin vs FN, H&E vs HE.

Response

Agree and thanks! We had made correction to address the non-uniformity.

6. There is no quantification of stainings in figure 3a-3d. further, why the jlp KO group was not included in WB and other assays? It would have been interesting to show the relative expression of jlp after KO and UUO.

Response

Thank you for your suggestion! We had added the quantification data and made comparison. The figure 3 we presented aims to addressing the expression profile of JLP and TGF- β 1 in the context of kidney fibrosis progression. So we had not included the jlp KO group here, but was laid in other figure to compare with other group.

7. Throughout the manuscript, “abundance” has been written as “abandance”.

Response

Thanks! We had fixed the mistake.

8. The sequence of results needs to be rearranged as the present sequence creates some confusion while reading. The subheading 3 can be made subheading 1. The authors are also advised to show the in vitro autophagy induction results first, followed by in vivo results to make it clearer.

Response

Thank you for your advice! We had made some adjust in the manuscript.

9. The quantification of stainings, IF, IHC should be more descriptive. What was the number of samples used? How many visual fields were considered? How the results were analysed?

Response

Thank you for your advice! We had provided more information in the manuscript.

10. Discussion part needs slight revision. The authors are advised to correlate their findings with earlier published reports.

Response

Thank you for your suggestion! We had cited the earlier report in the manuscript.

Minor comments: The manuscript is full of typographical and grammatical errors E.g.

11. “in particularly, in fibroblasts and myofibroblasts multiple”. Page 3

12. “examine the exression”. Page 8

13. expressed 26, with TNF- α , TGF- β 1 and FGF-2, which are the key initiators of

14. inflammatory and fibrotic lesions 3,72. Last paragraph, page 8

15. “of which are over-produced” page 9

16. The authors need to modify the sentence “whereas more Jlp was identified in the proximal tubules in wildtype and normal kidneys”. Page 15. As the expression of jlp can also be observed in the distal tubules. However, the expression was more in the proximal tubule. The discussion needs to be modified where the authors are comparing the jlp with klotho and BMP-7.

17. EMT production or ECM production? Page 18

Response

Thanks a lot! The mistakes are all corrected in the manuscript.

18. Where was the chloroquine used? Was it used as standard autophagy inhibitor for comparison?

Response

That is a mistake to mention chloroquine, we had delete it.

19. The human kidney specimen collection should be more descriptive or mention it is provided in supplementary data.

Response

The description of JLP expression in human kidney specimen has been moved to the supplementary data.

20. Primer or primary pairs? Page 24, rtPCR

Response

Thanks a lot! The mistakes are all corrected in the manuscript.

21. Statistical analysis should be more descriptive in the figure legends. How the groups were compared?

Response

Thanks a lot! Statistical analysis has been described in detail in the revised manuscript.

22. Phosphor-smad3. Page 43

Response

Thanks a lot! The mistakes are all corrected in the manuscript.

23. Why was the staining for F4/80 used? How to correlate it with the findings of present study?

Response

Staining F4/80 is aimed to evaluate the inflammatory damage in the kidney fibrosis..

24. Date are presented as mean \pm SDs. Page 43. It should be data.

Response

Thanks a lot! The mistakes are all corrected in the manuscript.

25. Number 1 and 2 indicate each individual sample in each given group. There are no number 1 and 2. Page 45. Similarly, on page 47

Response

The mistakes you pointed out have now been corrected. Thanks!

REVIEWERS' COMMENTS:

Reviewer #1 (Remarks to the Author):

The authors have appropriately responded to the reviewers' comments, and successfully revised the manuscript according to them.

Reviewer #3 (Remarks to the Author):

The revised version of the manuscript addressed all the comments raised by me. Manuscript can be accepted